# Predify: Augmenting deep neural networks with brain-inspired predictive coding dynamics

**Bhavin Choksi**[*]
CerCo CNRS, UMR 5549 &
Université de Toulouse
bhavin.choksi@cnrs.fr

**Milad Mozafari**[*]
CerCo CNRS, UMR 5549 &
IRIT CNRS, UMR 5505
milad.mozafari@cnrs.fr

**Callum Biggs O'May**
CerCo CNRS
UMR 5549

**Benjamin Ador**
CerCo CNRS
UMR 5549

**Andrea Alamia**
CerCo CNRS
UMR 5549

**Rufin VanRullen**
CerCo CNRS, UMR 5549 &
ANITI, Université de Toulouse
rufin.vanrullen@cnrs.fr

## Abstract

Deep neural networks excel at image classification, but their performance is far less robust to input perturbations than human perception. In this work we explore whether this shortcoming may be partly addressed by incorporating brain-inspired recurrent dynamics in deep convolutional networks. We take inspiration from a popular framework in neuroscience: "predictive coding". At each layer of the hierarchical model, generative feedback "predicts" (i.e., reconstructs) the pattern of activity in the previous layer. The reconstruction errors are used to iteratively update the network's representations across timesteps, and to optimize the network's feedback weights over the natural image dataset–a form of unsupervised training. We show that implementing this strategy into two popular networks, VGG16 and EfficientNetB0, improves their robustness against various corruptions and adversarial attacks. We hypothesize that other feedforward networks could similarly benefit from the proposed framework. To promote research in this direction, we provide an open-sourced PyTorch-based package called *Predify*, which can be used to implement and investigate the impacts of the predictive coding dynamics in any convolutional neural network.

## 1 Introduction

Deep convolutional neural networks (DCNNs), initially inspired by the primate visual cortex architecture, have taken big strides in solving computer vision tasks in the last decade. State-of-the-art networks can learn to classify images with high accuracy from huge labeled datasets [1–6]. This rapid progress and the resulting interest in these techniques have also highlighted their various shortcomings. Most widely studied is the sensitivity of neural networks, not only to perturbations specifically designed to fool them (so-called "adversarial examples") but also to regular noises typically observed in natural scenes [7–9]. These shortcomings indicate that there is still room for improvement in current techniques.

One possible way to improve the robustness of artificial neural networks could be to take further inspiration from the brain. In particular, one major aspect of the cerebral cortex that is missing from standard feedforward DCNNs is the presence of feedback connections. Recent studies have stressed the importance of feedback connections in the brain [10, 11], and have shown how artificial neural

---

[*]Equal Contribution

35th Conference on Neural Information Processing Systems (NeurIPS 2021).

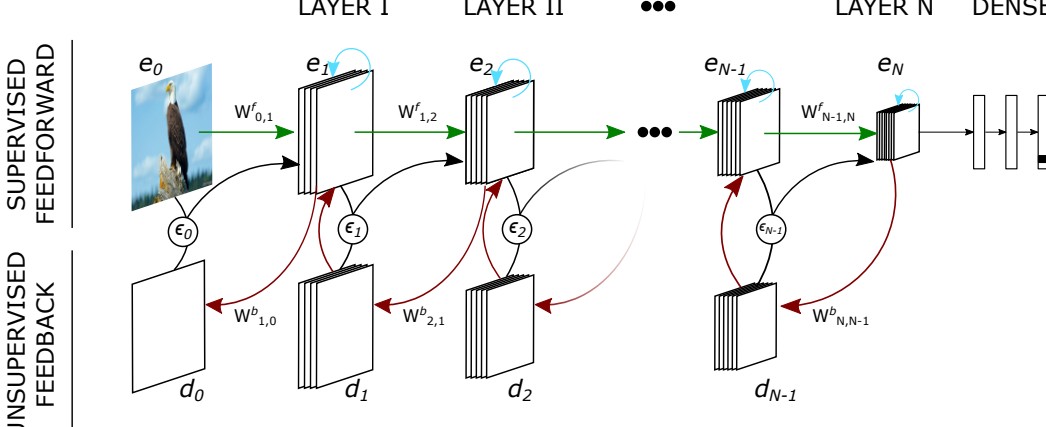

Figure 1: **General overview of our predictive coding strategy** as implemented in a feedforward hierarchical network with generative feedback connections. The architecture (roughly similar to stacked auto-encoders) consists of $N$ encoding layers $e_n$ and $N$ decoding layers $d_n$. $W_{m,n}$ denotes the connection weights from layer $m$ to layer $n$, with $W^f$ and $W^b$ for feedforward and feedback connections, respectively. The reconstruction errors at each layer are denoted by $\epsilon_n$. The feedforward connections (green arrows) are trained for image classification (in a supervised fashion), while the feedback weights (red arrows) are optimized for a prediction (i.e. reconstruction) objective (unsupervised). Predictive coding minimizes the reconstruction errors in each layer by updating activations in the next layer accordingly (black arrows). Self-connections (memory) are represented by blue arrows.

networks can take advantage of such feedback for various tasks such as object recognition with occlusion [12], or panoptic segmentation [13]. Feedback connections convey contextual information about the state of the higher layers down to the lower layers of the hierarchy; in this way, they can constrain lower layers to represent inputs in meaningful ways. In theory, this could make neural representations more robust to image degradation [14]. Merely including feedback in the pattern of connections, however, may not always be sufficient; rather, it should be combined with proper mechanistic principles.

To that end, we explore the potential of recurrent dynamics for augmenting deep neural networks with brain-inspired predictive coding (supported by ample neuroscience evidence [15–19]). We build large-scale hierarchical networks with both feedforward and feedback connections that can be trained using error backpropagation. Several prior studies have explored this interesting avenue of research [20–23], but with important differences with our approach (see Section 3). We demonstrate that our proposed method adds desirable properties to feedforward DCNNs, especially when viewed from the perspective of robustness.

Our contributions can be summarized as follows:

- We propose a novel strategy for effectively incorporating recurrent feedback connections based on the neuroscientific principle of predictive coding.

- We implement this strategy in two pre-trained feedforward architectures with unsupervised training of the feedback weights, and show that this improves their robustness against different types of natural and adversarial noise.

- We suggest and verify that an emergent property of the network is to iteratively shift noisy representations towards the corresponding clean representations—a form of "projection towards the learned manifold" as implemented in certain adversarial defense methods.

- To facilitate research aimed at using such neuroscientific principles in machine learning, we provide a Python package called *Predify* that can easily implement the proposed predictive coding dynamics in any convolutional neural network with a few lines of code.

## 2 Our Approach

### 2.1 The proposed predictive coding dynamics

Predictive coding, as introduced by [24], is a neurocomputational theory positing that the brain maintains an internal model of the world, which it uses to actively predict the observed stimulus. Within a hierarchical architecture, each higher layer attempts to predict the activity of the layer immediately below, and the errors made in this prediction are then utilized to correct the higher-layer activity.

To establish our notation, let us consider a hierarchical feedforward network equipped with generative feedback connections, as represented in Figure 1. The network contains $N$ encoding layers $e_n$ ($n \in \mathbb{N}$) and $N$ corresponding decoding layers $d_{n-1}$. The feedforward weights connecting layer $n-1$ to layer $n$ are denoted by $W_{n-1,n}^f$, and the feedback weights from layer $n+1$ to $n$ by $W_{n+1,n}^b$. For a given input image, we first initiate the activations of all encoding layers with a feedforward pass. Then, over successive recurrent iterations (referred to as timesteps $t$), both the decoding and encoding layer representations are updated using the following equations (also refer to Pseudocode 1):

$$\boldsymbol{d}_n(t) = W_{n+1,n}^b \boldsymbol{e}_{n+1}(t) \tag{1}$$

$$\boldsymbol{e}_n(t+1) = \beta_n W_{n-1,n}^f \boldsymbol{e}_{n-1}(t+1) + \lambda_n \boldsymbol{d}_n(t) + (1 - \beta_n - \lambda_n)\boldsymbol{e}_n(t) - \alpha_n \nabla \epsilon_{n-1}(t), \tag{2}$$

where $\beta_n$, $\lambda_n$ ($0 \leq \beta_n + \lambda_n \leq 1$), and $\alpha_n$ act as layer-dependent balancing coefficients for the feedforward, feedback, and error-correction terms, respectively. $\epsilon_{n-1}(t)$ denotes the reconstruction error at layer $n-1$ and is defined as the mean squared error (MSE) between the representation $e_{n-1}(t)$ and the predicted reconstruction $d_{n-1}(t)$ at that particular timestep. Layer $e_0$ is defined as the input image and remains constant over timesteps. All the weights $W_{n-1,n}^f$ and $W_{n+1,n}^b$ are fixed during these iterations.

Each of the four terms in Equation 2 contributes different signals, reflected by different arrow colors in Figure 1: (i) the feedforward term (green arrows; controlled by parameter $\beta$) provides information about the (constant) input and changing representations in the lower layers, (ii) the feedback correction term (red arrows; parameter $\lambda$), as proposed in [24, 25], guides activations towards their representations from the higher levels, thereby reducing the reconstruction errors over time, (iii) the memory term (blue arrows) acts as a time constant to retain the current representation over successive timesteps, and (iv) the feedforward error correction term (black arrows; controlled by parameter $\alpha$) corrects representations in each layer such that their generative feedback can better match the preceding layer. For this error correction term, we directly use the error gradient $\nabla \epsilon_{n-1} = [\frac{\partial \epsilon_{n-1}}{\partial e_n^0}, ..., \frac{\partial \epsilon_{n-1}}{\partial e_n^k}]$ to take full advantage of modern machine learning capabilities (where $k$ is the number of elements in $e_n$). While the direct computation of this error gradient is biologically implausible, it has been noted before that it is mathematically equivalent to propagating error residuals

---

**Pseudocode 1** Predictive Coding Iterations

---

1: Input image: $e_0$
2: **for** $n = 1$ to $N$ **do**
3:     $e_n \leftarrow Conv(e_{n-1})$
4:     $d_{n-1} \leftarrow deConv(e_n)$
5:     $\epsilon_{n-1} \leftarrow ||d_{n-1} - e_{n-1}||_2^2$
6: **end for**
7: **for** $t = 1$ to $T$ **do**
8:     **for** $n = 1$ to $N$ **do**
9:         $ff \leftarrow \beta_n \cdot Conv(e_{n-1})$
10:         $fb \leftarrow 0$
11:         **if** $n < N$ **then**
12:             $fb \leftarrow \lambda_n \cdot d_n$
13:         **end if**
14:         $e_n \leftarrow ff + fb + (1 - \beta_n - \lambda_n) \cdot e_n - \alpha_n \cdot \nabla \epsilon_{n-1}$
15:         $d_{n-1} \leftarrow deConv(e_n)$
16:         $\epsilon_{n-1} \leftarrow ||d_{n-1} - e_{n-1}||_2^2$
17:     **end for**
18: **end for**

---

up through the (transposed) feedback connection weights $(W^b)^T$, as often done in other predictive coding implementations [22, 24]. Together, the feedforward and feedback error correction terms fulfill the objective of predictive coding as laid out by Rao and Ballard [24]. We discuss the similarities and differences between our equations and those proposed in the original Rao and Ballard implementation in the Appendix A.6.

While it is certainly possible to train such an architecture in an end-to-end fashion, by combining a classification objective for the feedforward weights $W^f$ with an unsupervised predictive coding objective (see Section 2.2) for the feedback weights $W^b$, we believe that the benefits of our proposed scheme are best demonstrated by focusing on the added value of the feedback pathway onto a pre-existing state-of-the-art feedforward network. Consequently, we implement the proposed strategy with two existing feedforward DCNN architectures as backbones: VGG16 and EfficientNetB0, both trained on ImageNet. We show that predictive coding confers higher robustness to these networks.

## 2.2   Model architectures and training

We select VGG16 and EfficientNetB0, two different pre-trained feedforward networks on ImageNet, and augment them with the proposed predictive coding dynamics. The resulting models are called PVGG16 and PEfficientNetB0, respectively. The networks' "bodies" (without the classification head) are split into a cascade of $N$ sub-modules, where each plays the role of an $e_n$ in equation (2). We then add deconvolutions as feedback layers $d_{n-1}$ connecting each $e_n$ to $e_{n-1}$, with kernel sizes accounting for the increased receptive fields of the neurons in $e_n$ or upsampling layers to match the size of the predictions and their targets (see Appendix A.2). We then train the parameters of the feedback deconvolution layers with an unsupervised reconstruction objective (with all feedforward parameters frozen). We minimize the reconstruction errors just after the first forward pass, and after a single deconvolution step (i.e. no error correction or predictive coding recurrent dynamics are involved at this stage):

$$\mathcal{L} = \sum_{n=0}^{N-1} \parallel e_n - d_n \parallel_2^2, \tag{3}$$

where $e_n$ is the output of the $n^{th}$ encoder after the first forward pass and $d_n$ is the estimated reconstruction of $e_n$ via feedback/deconvolution (from $e_{n+1}$).

For both the networks, after training the feedback deconvolution layers, we freeze all of the weights, and set the values of hyperparameters to $\beta_n = 0.8$, $\lambda_n = 0.1$, and $\alpha_n = 0.01$ for all the encoders/decoders in Equation (2). We also explore various strategies for further tuning hyperparameters to improve the results (see Appendix A.7 for the chosen hyperparameter values).

## 2.3   *Predify*

To facilitate and automate the process of adding the proposed predictive coding dynamics to existing deep neural networks, we have developed an open-source Python package called *Predify*. The package is developed based on PyTorch [26] and provides a flexible object oriented framework to convert any PyTorch-compatible network into a predictive network. While an advanced user may find it easy to integrate *Predify* in their project manually, a simple text-based user interface (in TOML[2] format) is also provided to automate the steps. For the sake of improved performance and flexibility, *Predify* generates the code of the predictive network rather than the Python object. Given the original network and a configuration file (e.g. `'config.toml'`) that indicates the intended source and target layers for the predictive feedback, three lines of code are enough to construct the corresponding predictive network:

```
from predify import predify

net = # load PyTorch network
predify(net,'./config.toml') # config file indicates the layers that
                             # will act as outputs of encoders.
```

---

[2] https://toml.io/en/

The Appendix A.1 provides further details on the package, along with a sample config file and certain default behaviours. *Predify* is an ongoing project available on GitHub[3] under GNU General Public License v3.0. Scripts for creating PVGG16 and PEfficientNetB0 from their feedforward instances and reproducing the results presented in this paper, as well as the pre-trained weights are also available on another GitHub repository[4].

## 3    Related work

There is a long tradition of drawing inspiration from neuroscience knowledge to improve machine learning performance. Some studies suggest using sparse coding, a concept closely related to predictive coding [16, 27–30], for image denoising [31] and robust deep learning [32, 33], while other studies focus on implementing feedback and horizontal recurrent pathways to tackle challenges beyond the core object recognition [13, 14, 25, 34–39].

Here, we focus specifically on those studies that tried implementing predictive coding mechanisms in machine learning models [20–23]. Out of these, our implementation is most similar to the Predictive Coding Networks (PCNs) of [22]. These hierarchical networks were designed with a similar goal in mind: improving object recognition with predictive coding dynamics. However, their network (including the feedback connection weights) is solely optimized with a classification objective. As a result, their network does not learn to uniformly reduce reconstruction errors over timesteps, as the predictive coding theory would mandate. We also found that their network performs relatively poorly until the final timestep (see corresponding Figure S1 in the Appendix A.5), which does not seem biologically plausible: biological systems typically cannot afford to wait until the last iteration before detecting a prey or a predator. In the proposed method, we incorporate the feedforward drive into a similar PC dynamics and train the feedback weights in an unsupervised way using a reconstruction loss. We then show that these modifications help resolve PCNs' issues. We discuss these PCNs [22] further in the Appendix A.5, together with our own detailed exploration of their network's behavior.

Other approaches to predictive coding for object recognition include Boutin et al. [23], who used a PCN with an additional sparsity constraint. The authors showed that their framework can give rise to receptive fields which resemble those of neurons in areas V1 and V2 of the primate brain. They also demonstrated the robustness of the system to noisy inputs, but only in the context of reconstruction. Unlike ours, they did not show that their network can perform (robust) classification, and they did not extend their approach to deep neural networks.

Spratling [40] also described PCNs designed for object recognition, and demonstrated that their network could effectively recognise digits and faces, and locate cars within images. Their update equations differed from ours in a number of ways: they used divisive/multiplicative error correction (rather than additive), and a form of biased competition to make the neurons "compete" in their explanatory power. The weights of the network were not trained by error backpropagation, making it difficult to scale it to address modern machine learning problems. Conversely, our proposed network architecture and PC dynamics are fully compatible with error backpropagation, making them a suitable option for large-scale problems. Indeed, the tasks on which they tested their network are simpler than ours, and the datasets are much smaller.

Huang et al. [41] also aimed to extend the principle of predictive coding by incorporating feedback connections such that the network maximizes "self consistency" between the input image features, latent variables and label distribution. The iterative dynamics they proposed, though different from ours, improved the robustness of neural networks against gradient-based adversarial attacks on datasets such as Fashion-MNIST and CIFAR10.

## 4    Results

Here we contrast the behavior of feedforward networks with their predictive coding augmentations. When considered at timestep 0 (i.e., after a single feedforward and feedback pass through the model), the deep predictive coding networks (DPCNs) and their accuracy are—by construction—exactly identical to their standard pretrained feedforward versions. Over successive timesteps, however,

---

[3]`https://github.com/miladmozafari/predify`
[4]`https://github.com/bhavinc/predify2021`

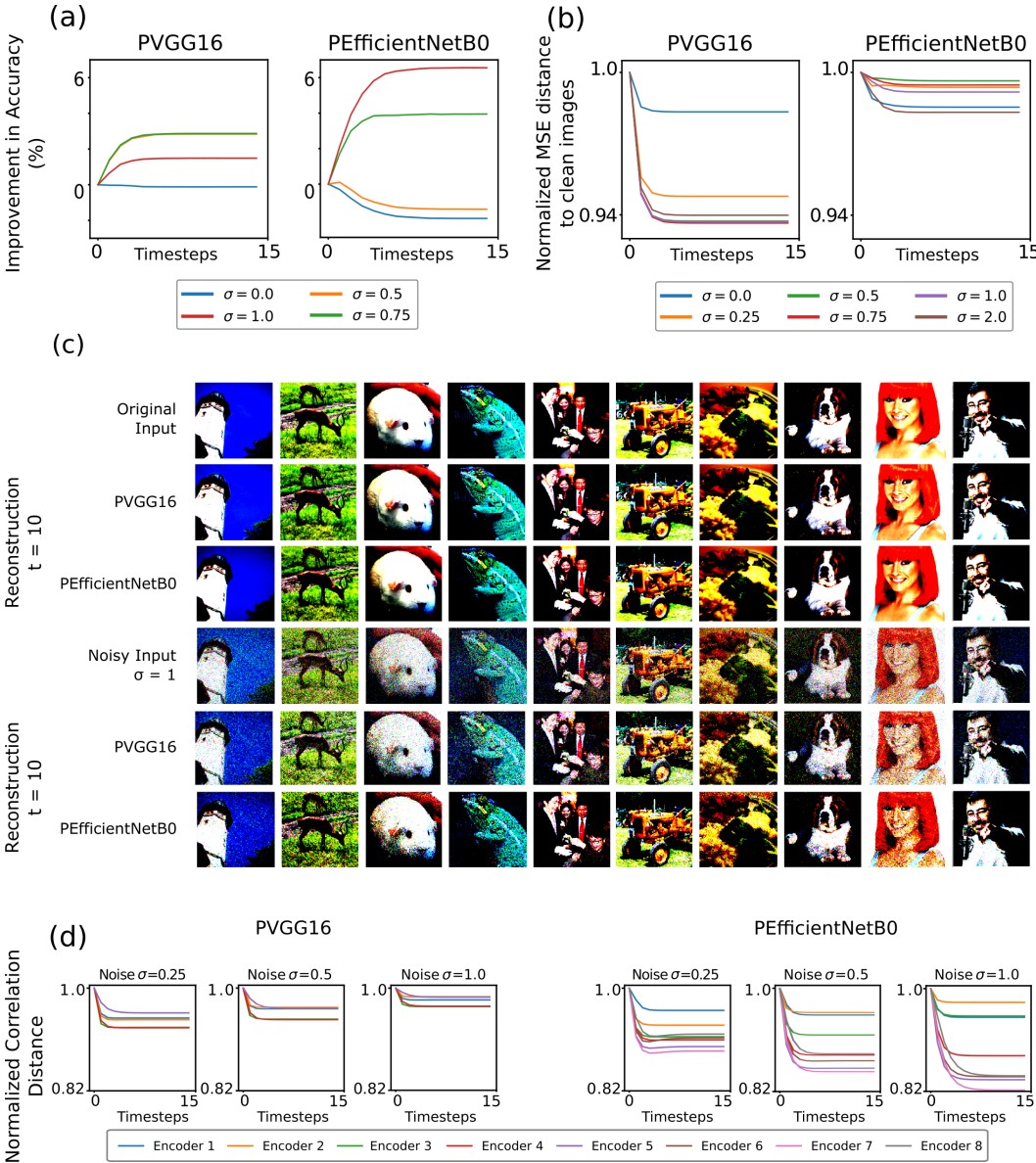

Figure 2: **Performance under Gaussian noise and projection towards the learned manifold.** (a) Improvement in recognition accuracy with reference to the feedforward baseline under various levels of Gaussian noise. Both networks demonstrate significant accuracy improvement across timesteps under noisy conditions, while maintaining a performance close to the feedforward level for clean images. (b) Normalized MSE distance between the image reconstruction ($d_0$) and the clean image ($e_0$). Irrespective of the noise level, image reconstruction consistently gets closer to the clean image across timesteps in both models. (c) Examples of clean and noisy input images together with their final reconstruction by the model (the row order from top to bottom is: original image, PVGG16 reconstruction, PEfficientNetB0 reconstruction; noisy image, PVGG16 reconstruction, PEfficientNetB0 reconstruction). For best viewing, we recommend zooming in on the electronic version. (d) Normalized correlation distance between representation of clean and noisy images for each encoder ($e_i$) across timesteps. The values are normalized with respect to the feedforward baseline (timestep 0). In both models and all encoders, the noisy representations tend to move toward the clean copies.

the influence of feedback and predictive coding iterations becomes visible. Here, we investigate for both DPCNs (PVGG16 and PEfficientNetB0): (i) how the PC dynamics update the networks' representations across timesteps, and in which direction relative to the learned manifold; (ii) how the networks benefit from PC under noisy conditions, or against adversarial attacks.

## 4.1 Performance under Gaussian noise

To understand the evolution of representations and the behavior of the proposed DPCNs, we first investigate their performance under the influence of different levels of Gaussian noise. To this end, we inject additive Gaussian noise to the ImageNet validation set, and monitor the models' performance across timesteps.

In Figure 2a we provide the classification accuracy on these noisy images and absolute values in the Table S4. We observed that both models progressively improve their recognition accuracy relative to their feedforward baseline (timestep 0) over successive iterations while imposing only a minor performance reduction on clean images. In other words, the networks are able to discard some of the noise by leveraging the predictive coding dynamics over timesteps.

## 4.2 Projection towards the learned manifold

In order to quantify DPCNs' denoising ability, we evaluate the quality of image reconstructions generated by each network using the mean squared error (MSE) between the clean image and its reconstruction generated by the first decoder. For each DPCN, we normalize these distances, by dividing them by the value obtained for the corresponding feedforward network (at t=0). We provide the absolute values in the Table S5. As Figures 2b-c illustrate, the reconstructions become progressively cleaner over timesteps. It should be noted that the feedback connections were trained only to reconstruct clean images; therefore, this denoising property is an emerging feature of the PC dynamics.

Next, we test whether the higher layers of the proposed DPCNs also manifest this denoising property. Hence, we pass clean and noisy versions of all images from the ImageNet validation set through the networks, and measure the average correlation distance between the clean and noisy representations of each encoder at each timestep. As done above, these correlation distances are then normalized with the distance measured at timestep 0 (i.e., relative to the standard feedforward network). For both the networks, the correlation distances decrease consistently over timesteps across all layers (see Figure 2d). This implies that predictive coding iterations help the networks to steer the noisy representations closer to the representations elicited by the corresponding (unseen) clean image.

This is an important property for robustness. When compared to clean images, noisy images can result in different representations at higher layers [42] and consequently, produce significant classification errors. Various defenses have aimed to protect neural networks from perturbations and adversarial attacks by constraining the images to the "original data manifold". Accordingly, studies have used generative models such as GANs [43–46] or PixelCNNs [47] to constrain the input to the data manifold. Similarly, multiple efforts have been made to clean the representations in higher layers and keep them closer to the learned latent space [42, 48–50]. Here, we demonstrate that feedback predictive coding iterations can achieve a similar goal by iteratively projecting noisy representations towards the manifolds learned during training, both in pixel (Figure 2b-c) and representation spaces (Figure 2d).

## 4.3 Benchmarking robustness to ImageNet-C

Given the promising results with additive Gaussian noise (Figure 2), we extend the noise variety and quantify the classification accuracy of the networks under different types of perturbations. We use ImageNet-C, a benchmarking dataset for noise robustness provided by [8], including 19 types of image corruptions across 5 severity levels each. To begin with, we evaluate DPCNs with pre-defined hyperparameter values (as provided in subsection 2.2). We observe that they improve the Corruption Error (CE) scores over timesteps for several of the additive-noise corruptions: Gaussian noise, shot noise, impulse noise or speckle noise (see Figure 3), but fail to improve the overall mean Corruption Error, or mCE score (the recommended score for this benchmark [8]).

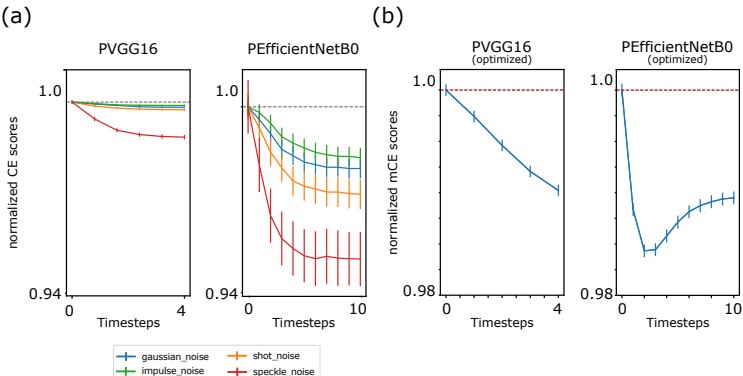

Figure 3: **Benchmarking robustness to ImageNet-C.** (a) Normalized corruption errors (CE) of PVGG16 and PEfficientNetB0 under four types of additive noise corruptions. The values are normalized with respect to the feedforward baseline. Both networks show consistent reductions in the errors across timesteps. (b) Normalized mean Corruption Error (mCE) scores for PVGG16 and PEfficientNetB0 on all the 19 corruptions available in the ImageNet-C dataset, when optimized hyperparameters are used (as described in the Appendix A.7). The values are normalized with respect to the feedforward baseline. In both the panels, error bars represent the standard deviation of the bootstrapped estimate of the mean value.

Thus, instead of using pre-defined hyperparameter values, we fine-tune them using two different methods (see Appendix A.7), and repeat the above experiment. As shown in Figure 3b, when the hyperparameters are more appropriately tuned for the task, the PC dynamics can increase noise robustness more generally across noise types, resulting in improvements of the mean Corruption Error (mCE) score. The CE plots for individual perturbations along with other recommended metrics (values normalized with AlexNet scores, Relative mCE scores) are provided in the Appendix A.8.

Furthermore, in the Appendix A.9, we demonstrate that we can replicate these observations with a version of PEfficientNetB0 provided by [51] that is robust to corruptions in the ImageNet-C dataset. We show that the recurrent dynamics we propose still help in further improving the mCE score of this already robust network.

### 4.4 Benchmarking robustness to adversarial attacks

Finally, we evaluate the robustness of the networks across timesteps against adversarial attacks. The proposed DPCNs are recurrent models, meaning that their layer representations change on every timestep, and consequently, so do the classification boundaries in the last layer, leading to different accuracy and generalization errors across time (as seen above). To mitigate this effect and properly assess the changes in robustness due to the PC dynamics, for each network we start by selecting 1000 images from the ImageNet validation dataset such that they are correctly classified across all timesteps. Also, we only perform *targeted* attacks so that for each image, the same attack target is given for all timesteps. Using the *Foolbox* library [52], we conduct targeted Basic_Iterative_Method attacks (BIM, with $L_\infty$ norm) [53] for both networks; although it would prove computationally prohibitive to systematically explore all standard types of adversarial attacks, we also evaluated random Projected Gradient Descent attacks (RPGD, with $L_2$ norm) [54], and non-gradient-based HopSkipJump attacks [55] on a subset of 100 images, specifically for PEfficientNetB0. Across various levels of allowed image perturbations (denoted as $\epsilon$s), the predictive coding iterations tend to decrease the success rate of the attacks across timesteps, for both networks and attacks (see Figure 4). That is, DPCNs are more robust against these adversarial attacks than their feedforward counterparts.

## 5 Discussion and Conclusion

In this work, we explore the use of unsupervised recurrent predictive coding (PC) dynamics, based on neuroscientific principles, to augment modern deep neural networks. The resulting models have an initial feedforward sweep, compatible with visual processing in human and macaque brains [11, 56–58]. Following this feedforward sweep, consecutive layers iteratively exchange information regarding

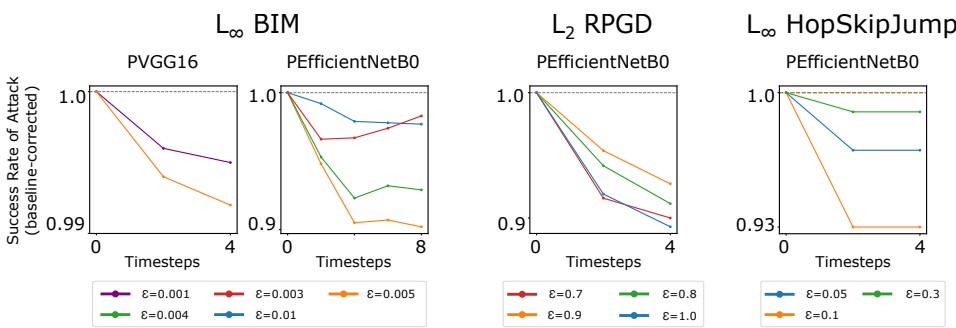

Figure 4: **Benchmarking robustness to adversarial attacks.** Plots show the success rate of targeted adversarial attacks against DPCNs across timesteps. The values are baseline-corrected, relative to the success rate at timestep 0 (feedforward baseline). Though we see some signs of reversals for a few perturbations, on average, both networks demonstrate improving robustness across timesteps to different types and/or levels of perturbations.

predictions and prediction errors, aiming to converge towards a stable explanation of the input. This dynamic system is inspired by, and reminiscent of, the "canonical microcircuit" (a central component of cortical structure [15]) that relies on feedback signaling between hierarchically adjacent layers to update its activity. Overall, the augmented networks are closer to the architecture of biological visual systems, while gaining some desirable functional properties. For example, in [59], we also demonstrated that the proposed dynamics help the networks perceive illusory contours in a similar way to humans.

Here, we implemented these PC dynamics in two state-of-the-art DCNs, VGG16 and EfficientNetB0, and showed that they helped to improve the robustness of the networks against various corruptions (e.g. ImageNet-C). We demonstrated that this behavior, at least partly, stems from PC's ability to project both the corrupted image reconstructions and neural representations towards their clean counterparts.

We also tested the impact of our network augmentations against adversarial attacks; here again, we showed that PC helps to improve the robustness of the networks. So far, the most promising strategy for achieving robustness has been adversarial training, whereby adversarial datapoints are added to the training dataset. While efficient, this strategy was also shown to be strongly limited [60, 61]. Apart from factors like the choice of the norms used for training, or the high computation requirements, it is ultimately performed with a supervised loss function that can alter the decision boundaries in undesirable ways [61, 62]. Most importantly, adversarial training shares very little, if any, resemblance to the way the brain achieves robustness. Instead, here we start from biological principles and show that they can lead to improved adversarial robustness. It is worth mentioning that both our networks achieved robustness totally via unsupervised training of the feedback connections (while of course, the backbone feed-forward networks that we used were pretrained in a supervised manner). We avoided using costly adversarial training, or tuning our hyperparameters specifically for classification under each attack. This likely explains why the models, while improving in robustness compared to their feedforward versions, remain far from state-of-the-art adversarial defenses. On the other hand, we believe that addition of these methods (adversarial training, hyperparameter tuning) to the training paradigm, in future work, could further improve the networks' adversarial robustness.

For the present experiments, we made a choice of using different objectives for training the feedforward and feedback weights: pre-trained feedforward weights optimized for classification, feedback weights trained with a reconstruction objective (computed after a single time-step). On the one hand, we note that it is perfectly feasible to train a similar predictive coding architecture with a single objective (classification, reconstruction, or otherwise) for both feedforward and feedback weights [59, 63]. On the other hand, our choice has several advantages. First, using a feedforward backbone pretrained for classification allowed us to demonstrate the effect of our dynamics on pre-existing state-of-the-art neural networks. Some authors have tried training both feedforward and feedback connections together for classification [22] at the final timestep for relatively smaller networks, but as we discussed in our explorations in the Appendix A.5, we found that the resulting network ended up classifying correctly at the last timestep, with very poor performance during early timesteps. This problem could

be addressed by training over time-averaged metrics, such as the average cross-entropy loss for $N$ timesteps. Nonetheless, training the feedback weights for reconstruction instead of classification has the additional advantage that it can be done entirely without supervision. We chose to train the feedback weights for a single time-step, because training with recurrence over multiple timesteps would have required unrolling the network over time. Hence, training a large network like PVGG16 for say 5 or 10 timesteps would incur significant computational challenges. Furthermore, our use of a one-step reconstruction objective allowed us to train the feedback weights independently of the various hyperparameters of our predictive coding dynamics ($\beta$, $\lambda$, and $\alpha$), which only influence the model behavior after the second timestep. Training these weights using recurrence would have required to (i) either fix the values of these hyperparameters beforehand, leading to constraints of expensive hyperparameter explorations; (ii) or directly train these hyperparameters as parameters of the model, probably with additional constraints to prevent the network from reaching trivial values (e.g., if all hyperparameters but the feedforward term $\beta$ converge to zero, the network performs identically to a feedforward one). Finally, from a neuroscience perspective, whether and how the brain combines discriminative and generative representations has been an open question addressed by many researchers, e.g. [5, 64, 65]. Our approach of a discriminative (classification-trained) feedforward coupled with generative (reconstruction-trained) feedback could be considered another attempt in this direction.

We speculate that the proposed PC dynamics could help improve robustness in most feedforward neural architectures. To facilitate further explorations in this direction, we provided a Python package, called *Predify*, which allows users to implement recurrent PC dynamics in any feedforward DCN, with only a few lines of code. *Predify* automates the network building, and thus simplifies experiments. On the other hand, there is as yet no established method or criteria to automate the process of identifying the appropriate number of encoding layers, their source and target layers in the DCN hierarchy, and the corresponding hyperparameter values. This remains an open research question, and a requirement for manual explorations and tuning from *Predify* users. For instance, our own explorations with augmenting ResNets through *Predify* proved difficult, and failed in some situations but succeeded in others. More specifically, as developed in [63] using *Predify*, ResNet augmentations always achieved noise robustness when the hyperparameter values (controlling the feedforward, feedback, and memory terms) could be tuned separately for each noise type; but we found it challenging to identify a single set of hyperparameters that could generalize to all noise types. Nonetheless, we are hopeful that the package will prove useful to the community. The code is structured such that users can readily adapt it to test their hypotheses. In particular, it should allow both proponents and opponents of the predictive coding theory to investigate its effects on any DCN.

Overall, this work contributes to the general case for continuing to draw inspiration from biological visual systems in computer vision, both at the level of model architecture and dynamics. We believe that our user-friendly Python package *Predify* can open new opportunities, even for neuroscience researchers with little background in machine learning, to investigate bio-inspired hypotheses in deep computational models, and thus bridge the gap between the two communities.

## 6 Broader Impacts

The research discussed above proposes novel ways of using brain-inspired dynamics in current machine learning models. Specifically, it demonstrates a neuro-inspired method for improving the robustness of machine learning models. Given that such models are employed by the general public, and are simultaneously shown to be heavily vulnerable, research efforts to increase (even marginally) or to understand their robustness against mal-intentioned adversaries has high societal relevance.

Importantly, the research also aims to bridge techniques between two different fields–neuroscience and machine learning, which can potentially open new avenues for studying the human brain. For example, it could help better understand the unexplained neural activities in patients, to improve their living conditions, and in the best case, in the treatment of their conditions. While this may also be associated with inherent risks (related to privacy or otherwise), there are clear potential benefits to society.

The likelihood of sentient AI arising from this line of research is estimated to be rather low.

## Acknowledgments and Disclosure of Funding

This work was funded by an ANITI (Artificial and Natural Intelligence Toulouse Institute) Research Chair to RV (ANR grant ANR-19-PI3A-0004), as well as ANR grants AI-REPS (ANR-18-CE37-0007-01) and OSCI-DEEP (ANR-19-NEUC-0004).

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
