# A  Appendix

## A.1  Getting Started with Predify

Both VGG16 and EfficientNetB0 are converted to predictive coding networks PVGG16 and PEfficientNetB0, using the Predify package. The fastest and easiest way to convert a feedforward network into its predictive coding version is to use Predify's text-based interface which supports configuration files in TOML format.

The current version of Predify assumes that there is no gap between the encoders. Therefore, in the minimal case, one only needs to provide a list of sub-module names in the target feedforward network. Then, Predify takes care of the rest by converting each of them into an encoder and assigning default decoders. More precisely, let $x$ and $y$ denote the input and output of a layer (or complex sub-module, potentially including multiple layers) that is selected to be an encoder ($e_n$). If $x$ and $y$ respectively have the size $(c_{in}, h_{in}, w_{in})$ and $(c_{out}, h_{out}, w_{out})$; then, the default decoder's structure that predicts this encoder ($d_{n+1}$) is a 2D upscaling operation by the factor of $(h_{in}/h_{out}, w_{in}/w_{out})$ followed by a transposed convolutional layer with $c_{out}$ channels and $3 \times 3$ window size. The values of hyperparameters will be set to $\beta_n = 0.3$, $\lambda_n = 0.3$, and $\alpha_n = 0.01$

In Predify, each encoder ($e_n$) and the decoder that uses its output to predict the activity of the encoder below ($d_{n-1}$) is called a *PCoder*. To verify the functionality of Predify's default settings, we applied it for PEfficientNetB0 used in this work. Here is the corresponding minimal configuration file:

```
name = "PEfficientNetB0"

input_size = [3,224,224]
gradient_scaling = true
shared_hyperparameters = false

[[pcoders]]
module = "act1"
[[pcoders]]
module = "blocks[0]"
[[pcoders]]
module = "blocks[1]"
[[pcoders]]
module = "blocks[2]"
[[pcoders]]
module = "blocks[3]"
[[pcoders]]
module = "blocks[4]"
[[pcoders]]
module = "blocks[5]"
[[pcoders]]
module = "blocks[6]"
```

One can easily override the default setting by providing all the details for a PCoder. Here is the configuration corresponding to the PVGG16 used in this work:

```
imports = [
"from torch.nn import Sequential, ReLU, ConvTranspose2d",
]

name = "PVGG16"

input_size = [3, 224, 224]
gradient_scaling = true
shared_hyperparameters = false

[[pcoders]]
module = "features[3]"
predictor = "ConvTranspose2d(64, 3, kernel_size=(5, 5), stride=(1, 1),
    padding=(2, 2))"
hyperparameters = {feedforward=0.2, feedback=0.05, pc=0.02}
```

```
[[pcoders]]
module = "features[8]"
predictor = "Sequential(ConvTranspose2d(128, 64, kernel_size=(10, 10),
    stride=(2, 2), padding=(4, 4)), ReLU(inplace=True))"
hyperparameters = {feedforward=0.4, feedback=0.1, pc=0.05}

[[pcoders]]
module = "features[15]"
predictor = "Sequential(ConvTranspose2d(256, 128, kernel_size=(14, 14)
    , stride=(2, 2), padding=(6, 6)), ReLU(inplace=True))"
hyperparameters = {feedforward=0.4, feedback=0.1, pc=0.008}

[[pcoders]]
module = "features[22]"
predictor = "Sequential(ConvTranspose2d(512, 256, kernel_size=(14, 14)
    , stride=(2, 2), padding=(6, 6)), ReLU(inplace=True))"
hyperparameters = {feedforward=0.5, feedback=0.1, pc=0.0024}

[[pcoders]]
module = "features[29]"
predictor = "Sequential(ConvTranspose2d(512, 512, kernel_size=(14, 14)
    , stride=(2, 2), padding=(6, 6)), ReLU(inplace=True))"
hyperparameters = {feedforward=0.6, feedback=0.0, pc=0.006}
```

The network configuration files (in TOML format) are available to download on GitHub[5].

## A.2   Network Architectures

VGG16 consists of five convolution blocks and a classification head. Each convolution block contains two or three convolution+ReLU layers with a max-pooling layer on top. For each $e_n$ in PVGG16, we selected the max-pooling layer in block $n-1$ and all the convolution layers in block $n$ of VGG16 (for $n \in \{1, 2, 3, 4, 5\}$) as the sub-module that provides the feedforward drive. Afterwards, to predict the activity of each $e_n$, a deconvolution layer $d_n$ is added which takes the $e_{n+1}$ as the input. Here, deconvolution kernel sizes are set by taking the increasing receptive field sizes into account.

In the case of PEfficientNetB0, we used PyTorch implementation of EfficientNetB0 provided in `https://github.com/rwightman/pytorch-image-models`. This implementation of Efficient-NetB0 consists of eight blocks of layers (considering the first convolution and batch normalization layers as a separate block). Similar to PVGG16, we convert each of these blocks into an encoder ($e_n$) and add deconvolution layers accordingly. This time we set the kernel size of all deconvolution layers to 3x3 and use upsampling layers to compensate the shrinkage of layer size through the feedforward pathway (i.e. Predify's default setting).

Table S1 summarizes PVGG16's architecture. Moreover, the hyperparameter values are provided in Tables S2 and S3.

---

[5]`https://github.com/bhavinc/predify2021`

Table S1: Architectures of $e_n$s and $d_n$s for PVGG16 and PEfficientNetB0. Conv (channel, size, stride), MaxPool (size, stride), Deconv (channel, size, stride), Upsample (scale_factor), BN is BatchNorm, $[\,]_+$ is ReLU, and $[\,]_*$ is SiLU. EfficientBlock corresponds to each block in PyTorch implementation of EfficientNetB0.

| | PVGG16 Input Size: 3x224x224 | | PEfficientNetB0 Input Size: 3x224x224 | |
|---|---|---|---|---|
| | $e_n$ | $d_{n-1}$ | $e_n$ | $d_{n-1}$ |
| PCoder1 | $[\text{Conv }(64, 3, 1)]_+$ $[\text{Conv }(64, 3, 1)]_+$ | Deconv (3, 5, 1) | $[\text{BN (Conv }(32, 3, 2))]_*$ | Upsample (2) Deconv (3, 3, 1) |
| PCoder2 | MaxPool (2, 2) $[\text{Conv }(128, 3, 1)]_+$ $[\text{Conv }(128, 3, 1)]_+$ | $[\text{Deconv }(64, 10, 2)]_+$ | EfficientBlock0 | Deconv (32, 3, 1) |
| PCoder3 | MaxPool (2, 2) $[\text{Conv }(256, 3, 1)]_+$ $[\text{Conv }(256, 3, 1)]_+$ $[\text{Conv }(256, 3, 1)]_+$ | $[\text{Deconv }(128, 14, 2)]_+$ | EfficientBlock1 | Upsample (2) Deconv (16, 3, 1) |
| PCoder4 | MaxPool (2, 2) $[\text{Conv }(512, 3, 1)]_+$ $[\text{Conv }(512, 3, 1)]_+$ $[\text{Conv }(512, 3, 1)]_+$ | $[\text{Deconv }(256, 14, 2)]_+$ | EfficientBlock2 | Upsample (2) Deconv (24, 3, 1) |
| PCoder5 | MaxPool (2, 2) $[\text{Conv }(512, 3, 1)]_+$ $[\text{Conv }(512, 3, 1)]_+$ $[\text{Conv }(512, 3, 1)]_+$ | $[\text{Deconv }(512, 14, 2)]_+$ | EfficientBlock3 | Upsample (2) Deconv (40, 3, 1) |
| PCoder6 | - | - | EfficientBlock4 | Deconv (80, 3, 1) |
| PCoder7 | - | - | EfficientBlock5 | Upsample (2) Deconv (112, 3, 1) |
| PCoder8 | - | - | EfficientBlock6 | Deconv (192, 3, 1) |

## A.3 Execution Time

Since we used a variable number of GPUs for the different experiments, an exact execution time is hard to pinpoint. Briefly, depending on the number of timesteps, analysing mCE scores and adversarial attacks on PEfficientNetB0 took around 15-20 hours on an NVIDIA TitanV gpu. These numbers were about three to four times higher for experiments on PVGG16. For both the networks, training the feedback weights on the ImageNet dataset generally finished before 5 epochs, which took approximately 7-8 hours for a single GPU.

## A.4 Gradient Scaling

In our dynamics, the error ($\epsilon_{n-1}$) is defined as a scalar quantity whose gradient is taken with respect to the activation of the higher layer ($e_n$). That is,

$$\nabla\epsilon_{n-1} = \begin{bmatrix} \frac{\partial\epsilon_{n-1}}{\partial e_n^1} \\ \vdots \\ \frac{\partial\epsilon_{n-1}}{\partial e_n^L} \end{bmatrix} \tag{4}$$

where L denotes the number of elements in $e_n$. The partial derivative with respect to $e_n^j$ can then be written as,

$$\frac{\partial\epsilon_{n-1}}{\partial e_n^j} = \frac{1}{K}\sum_i^K \frac{\partial(e_{n-1}^i - d_{n-1}^i)^2}{\partial e_n^j} \tag{5}$$

$$\tag{6}$$

where K is the number of elements in $e_{n-1}$ ( = channels x width x height). Equation 5 highlights how the dimensionality of the prediction (equivalently the error term) affects the gradients, scaling them down by a factor K.

This can be easily seen by supposing that the gradients with respect to $e_n^j$ are i.i.d normally distributed around 0 with standard deviation $\sigma$,

$$\frac{\partial(e_{n-1}^i - d_{n-1}^i)^2}{\partial e_n^j} \sim \mathcal{N}(0, \sigma^2) \tag{7}$$

$$\sum_i^K \frac{\partial(e_{n-1}^i - d_{n-1}^i)^2}{\partial e_n^j} \sim \mathcal{N}(0, K\sigma^2) \tag{8}$$

Thus,

$$\frac{\partial \epsilon_{n-1}}{\partial e_n^j} = \frac{1}{K}\sum_i^K \frac{\partial(e_{n-1}^i - d_{n-1}^i)^2}{\partial e_n^j} \sim \mathcal{N}(0, \frac{\sigma^2}{K}) \tag{9}$$

This scaling is further troublesome in DCNs, where most gradients are zero since they are not part of the receptive field of the element $e_n^j$. Hence assuming that there are only C elements (kernel*channels) that are part of the receptive field of $e_n^j$,

$$\sum_i^K \frac{\partial(e_{n-1}^i - d_{n-1}^i)^2}{\partial e_n^j} = \sum_i^C \frac{\partial(e_{n-1}^i - d_{n-1}^i)^2}{\partial e_n^j} \sim \mathcal{N}(0, C\sigma^2) \tag{10}$$

Hence,

$$\frac{\partial \epsilon_{n-1}}{\partial e_n^j} = \frac{1}{K}\sum_i^C \frac{\partial(e_{n-1}^i - d_{n-1}^i)^2}{\partial e_n^j} \sim \mathcal{N}(0, \frac{C\sigma^2}{K^2}) \tag{11}$$

We use Equation 11 to, at least partly, counteract this effect due to the dimensionality of the prediction errors. We multiply the gradient by a factor of $\sqrt{K^2/C}$ to scale them in a way that is more comparable across layers, and thus apply a more meaningful step size for correcting the errors.

## A.5   Prior work: PCNs

To better understand the model proposed by Wen et al. [22] and its differences to ours, we conducted several experiments using the code that they provided, and report here our most compelling observations. A first striking shortcoming was that the accuracy of their feedforward baseline was far from optimal. Using their code, with relatively minor tweaks to the learning rate schedule, we were able to bring it up from ~60% to 70% – just a few percentage points below their recurrent network. We expect that this could be further improved with a more extensive and systematic hyperparameter search. In other words, their training hyperparameters appeared to have been optimised for their predictive coding network, but not – or not as much – for their feedforward baseline. We further found that a minor change to the architecture - using group normalisation layers after each ReLU – leads to a feedforward network which performs on par with the recurrent network, with a mean over 6 runs of 72% and best of 73%. Adding the same layers to the recurrent network did not lead to a corresponding improvement in accuracy.

We also found that the network had poor accuracy (underperforming the optimized feedforward baseline) until the final timestep, as can been seen in Figure S1b. This can be clarified by a closer reading of Figure 3 of their paper: the reported improvements over cycles from 60% at timestep 0 to more than 70% at timestep 6 are for seven distinct networks, each evaluated only at the timestep they were trained for. So in fact, in their model the predictive coding updates do not gradually improve on an already reasonably guess. This is clearly not biologically plausible: visual processing would be virtually useless if the correct interpretation of a scene only crystallised after a number of "timesteps". By the time a person has identified an object that object is likely to have disappeared or, in a worst

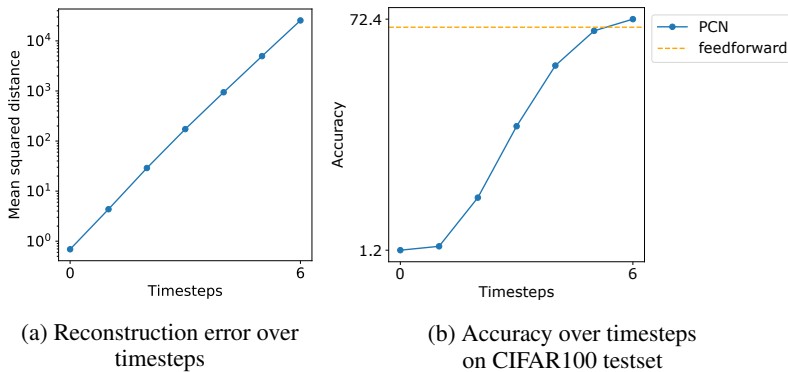

(a) Reconstruction error over timesteps

(b) Accuracy over timesteps on CIFAR100 testset

Figure S1: **PCN**: Panel (a) shows the reconstruction errors of the model over timesteps. It does not decrease over timesteps, as would be expected in a predictive coding system. Panel (b) depicts the accuracy of the model on the CIFAR100 test dataset. The model performs at chance level at early timesteps and then becomes better in the last few timesteps.

case scenario, eaten them. We also experimented with feeding the classification error at each timestep into an aggregate loss function, but this lead to a network which, while performing well, essentially did not improve over timesteps.

Figure S1a shows that the network does not uniformly minimise reconstruction errors over time for all layers, and thus is not performing correct predictive coding updates. In fact the total reconstruction error (across all layers) increases exponentially over timesteps. There are a number of possible explanations for this. Firstly, in the case of the network with untied weights, the authors choose to make a strong assumption in the update equations (seen as the equivalence of their Equations 5 and 6): that the feedback weights can be assumed to be the transpose of the feedforward weights, i.e. $W^b = (W^f)^T$. They thus propagate the feedforward error through the feedforward weights. However, it might be that the network learns feedback weights which essentially invert the feedforward transformation as assumed, but this is not guaranteed, and nor is it explicitly motivated through the classification loss function. Indeed, because the network is not motivated to learn a representation at earlier timesteps which produces a good prediction, it does not necessarily need to learn the inverse transformation: it can instead learn some other transformation which, when applied with the update equations, leads the network to *end up* in the right place. That being said, this assumption is valid for the network with tied weights, and this network also does not uniformly reduce reconstruction error over timesteps. Possibly, the presence of ReLU non-linearities means that the forward convolution may still not be perfectly invertible by a transposed convolution. Finally, in line with this unexpected increase of reconstruction errors over time, we have also failed to extract good image reconstructions from the network as seen in Figure 5 of their paper, although in private communication the authors indicated that this was possible with some other form of normalisation.

In short, while the ideas put forward in [22] share similarities with our own, their exact implementation did not support the claims of the authors, and the question of whether predictive coding can benefit deep neural networks remained an open one. We hope that our approach detailed in the present study can help resolve this question.

### A.6 Comparing with Rao and Ballard

This section aims to start from the equations initially provided in Rao and Ballard [24] and compare them to ours. The parallels drawn will help to highlight the similarities and the differences between both the approaches.

Rao and Ballard consider a two-layer system, and start with the assumption that the brain possesses a set of internal causes, denoted as $\mathbf{r}$ (in matrix notation), that it uses to predict the visual stimulus, for example an input image $\mathbf{I}$, such that

$$\mathbf{I} \approx f(U\mathbf{r}) \tag{12}$$

where $f(.)$ is some nonlinear activation function. This $\mathbf{r}$ can be equalled to encoding layer $e_1$ in our equations, with $\mathbf{I}$ being the input image $e_0$ or its reconstruction $d_0$. $U$ here, represents the top-down weight matrix (equivalent to $W_{1,0}^b$) that helps to make a prediction about the input image. That is,

$$\mathbf{I} \approx f(U\mathbf{r}) \equiv e_0 \approx d_0 = W_{1,0}^b e_1 \tag{13}$$

In this two-layer hierarchical architecture, $\mathbf{r}$ itself is predicted by the higher layer $\mathbf{r}^h$ using the weight matrix $U^h$, equivalent to how $e_1$ is predicted by $e_2$ using $W_{2,1}^b$ in our model. This prediction denoted as $\mathbf{r}^{td}$ in Rao and Ballard's original implementation can be equalled to $d_1$ in our equations.

$$\mathbf{r}^{td} = f(U^h \mathbf{r}^h) \equiv d_1 = W_{2,1}^b e_2 \tag{14}$$

The errors made in making the predictions are defined, like ours, as the mean squared distance,

$$\epsilon_0 = (\mathbf{I} - f(U\mathbf{r}))^T(\mathbf{I} - f(U\mathbf{r})) \tag{15}$$

$$\epsilon_1 = (\mathbf{r} - \mathbf{r}^{td})^T(\mathbf{r} - \mathbf{r}^{td}) \tag{16}$$

Please note that differentiating the prediction error $\epsilon_0$ with respect to $\mathbf{r}$ (similar to taking the gradient of $\epsilon_{n-1}$ with respect to $e_n$ as done in our error-correction term) gives us,

$$\nabla \epsilon_{\mathbf{0}} = -2U^T \frac{\partial f}{\partial U\mathbf{r}}^T (\mathbf{I} - fU\mathbf{r}) \tag{17}$$

$$= -kU^T(\mathbf{I} - f(U\mathbf{r})) \tag{18}$$

which will be useful later.

As per the predictive coding theory, the brain tries both to learn parameters ($U$ and $U^h$) over a dataset of natural inputs, and tries to modify its neural activations ($\mathbf{r}$ and $\mathbf{r}^h$) over time given a particular input, in such a way as to minimize the total error $E$, defined as:

$$E = a \cdot \underbrace{(\mathbf{I} - f(U\mathbf{r}))^T(\mathbf{I} - f(U\mathbf{r}))}_{\epsilon_0} + b \cdot \underbrace{(\mathbf{r} - \mathbf{r}^{td})^T(\mathbf{r} - \mathbf{r}^{td})}_{\epsilon_1} \tag{19}$$

Here $a$ and $b$ act as constants that weigh the errors in this two-level hierarchichal network. Equation 19 is reflected as Equation 4 on Page 86 of the original paper [24]. The original implementation also contains terms that account for the prior probability distributions of $\mathbf{r}$ and $U$; these terms can be equated to regularization terms, and thus we omit them for the sake of simplicity.

Equation 19 represents the overall error, calculated as sum of the mean squared errors across the hierarchy of the network. It should be noted that we use this same objective function ($-E$) to train the feedback weights of our networks.

As stated above, the predictive coding dynamics aim to modify neural representations $\mathbf{r}$ so as to minimize the error $E$, i.e., differentiating the above equation:

$$\frac{d\mathbf{r}}{dt} = -\frac{\partial E}{\partial \mathbf{r}} = a \cdot U^T \frac{\partial f^T}{\partial U\mathbf{r}}(\mathbf{I} - f(U\mathbf{r})) + b \cdot (\mathbf{r}^{td} - \mathbf{r}) \tag{20}$$

Barring a regularization term, the above equation is equivalent to Equation 7 on page 86 of [24]. One can see that the first term in the RHS of equation 20 can be substituted with our error-correction term $\nabla\epsilon_0$ (see Eq. 18). Hence, Equation 20 after simultaneously expanding the LHS becomes,

$$\frac{\mathbf{r}(t + dt) - \mathbf{r}(t)}{dt} = -a_1 \cdot \nabla\epsilon_r(t) + b \cdot (\mathbf{r}^{td}(t) - \mathbf{r}(t)) \tag{21}$$

We use subscript $r$ for $\epsilon$ to emphasize that this error can be calculated at any level/stage $\mathbf{r}$ represents in a multi-layer hierarchical system, and is not restricted to just the first layer of the hierarchy. Similarly, the time resolution $dt$ can be equated to 1 timestep (of arbitrary duration) for simulations. Hence, rearranging the equation further,

$$\mathbf{r}(t + 1) = \underbrace{b \cdot \mathbf{r}^{td}(t)}_{feedback} + \underbrace{(1 - b)\mathbf{r}(t)}_{memory} - \underbrace{a_1 \nabla\epsilon_r(t)}_{error-correction} \tag{22}$$

In the above equation, the first term corresponds to our feedback term, the second term corresponds to our memory term and the last term corresponds to our feedforward error-correction term. That is, exchanging constants to match our notation:

$$\mathbf{r}(t + 1) = \underbrace{\boxed{\phantom{xxxxxx}}}_{feedforward} + \underbrace{\lambda \cdot \mathbf{r}^{td}(t)}_{feedback} + \underbrace{(1 - \lambda)\mathbf{r}(t)}_{memory} - \underbrace{a_1 \nabla\epsilon_r(t)}_{error-correction} \tag{23}$$

This can be directly compared to our main Equation 2.

Equation 23 also highlights the fact that our approach has an extra feedforward term that is not present in the original Rao and Ballard proposal. We believe that such a modification allows for rethinking the role of error-correction in network dynamics; where error-correction constituted the predominant mode of feed-forward communication in the Rao and Ballard implementation, it plays a more supporting role in our implementation, iteratively correcting the errors made by the feedforward convolutional layers. We empirically found that the feedforward term helped to improve the stability of the training. Interestingly, a common criticism of predictive coding lies in its inability to explain the dominance of feedforward brain activity compared to prediction error signals [17, 18]. We believe that our proposed implementation allows for a flexible modulation of these two terms, and thus systematic investigation of these factors–as done in [63].

From a practical perspective, we expect that our framework can be readily used by both proponents and opponents of the predictive coding theory. Setting the feedforward term $\beta$ equal to zero produces a pure predictive coding network as proposed in Rao and Ballard [24]. Alternatively, one can set the error-correction term $\alpha$ equal to zero to study a bidirectional network with feedback and feedforward drives, in the style of Heeger [25]. The framework has been implemented such that the basic update rule (as class Pcoder in the package) is easily adaptable, allowing one to try other complex interactions between these terms; for example, one could easily include multiplicative interactions between feedback and feedforward terms to emulate forms of biased competition (see [40, 66]).

## A.7 Tuning hyperparameters

In addition to the fixed set of hyperparameters used in our initial experiments (Figures 2, 3a and 4), we also experimented with optimizing our hyperparameters. To tune the hyperparameters for the models, we applied two different strategies for both the models–tuning hyperparameters for the whole network vs tuning hyperparameters for each pcoder separately. After a few initial explorations on clean images, we discovered that the hyperparameters dictate where the network dynamics converge, and consequently its performance for noisy situations. This effect is characterized and investigated thoroughly in [63]. Thus, in this study, we decide to use gaussian noise of standard deviation 0.5 to tune the hyperparameters and test it on all other types of noises from the ImageNet-C dataset.

For PVGG16, we start by fixing the value of alpha for each layer to zero and only search for $\beta_n$'s and $\lambda_n$'s. We calculate the average cross-entropy loss for 4 timesteps on 2000 images and use it as a metric for choosing the hyperparameters. The hyperparameters chosen are as follows :

Table S2: Values of the Hyperparameters

| $n$ | $\beta_n$ | $\lambda_n$ | $\alpha_n$ |
|---|---|---|---|
| 1 | 0.2 | 0.05 | 0.01 |
| 2 | 0.4 | 0.10 | 0.01 |
| 3 | 0.4 | 0.10 | 0.01 |
| 4 | 0.5 | 0.10 | 0.01 |
| 5 | 0.6 | 0.00 | 0.01 |

For PEfficientNetB0, we take a different approach. Instead of the whole network, we start by finetuning each pcoder using the same metric (average crossentropy for 4 timesteps) on 4050 images. We then combine all hyperparameters found for each pcoder. The hyperparameters chosen are as follows :

Table S3: Values of the Hyperparameters

| $n$ | $\beta_n$ | $\lambda_n$ | $\alpha_n$ |
|---|---|---|---|
| 1 | 0.77 | 0.08 | 0.01 |
| 2 | 0.76 | 0.11 | 0.01 |
| 3 | 0.83 | 0.03 | 0.01 |
| 4 | 0.94 | 0.01 | 0.01 |
| 5 | 0.73 | 0.25 | 0.01 |
| 6 | 0.81 | 0.01 | 0.01 |
| 7 | 0.85 | 0.10 | 0.01 |
| 8 | 1.0 | 0.00 | 0.01 |

We then, calculate the mCE scores using all the 19 noises for both the networks. The CE scores for each noise are shown below :

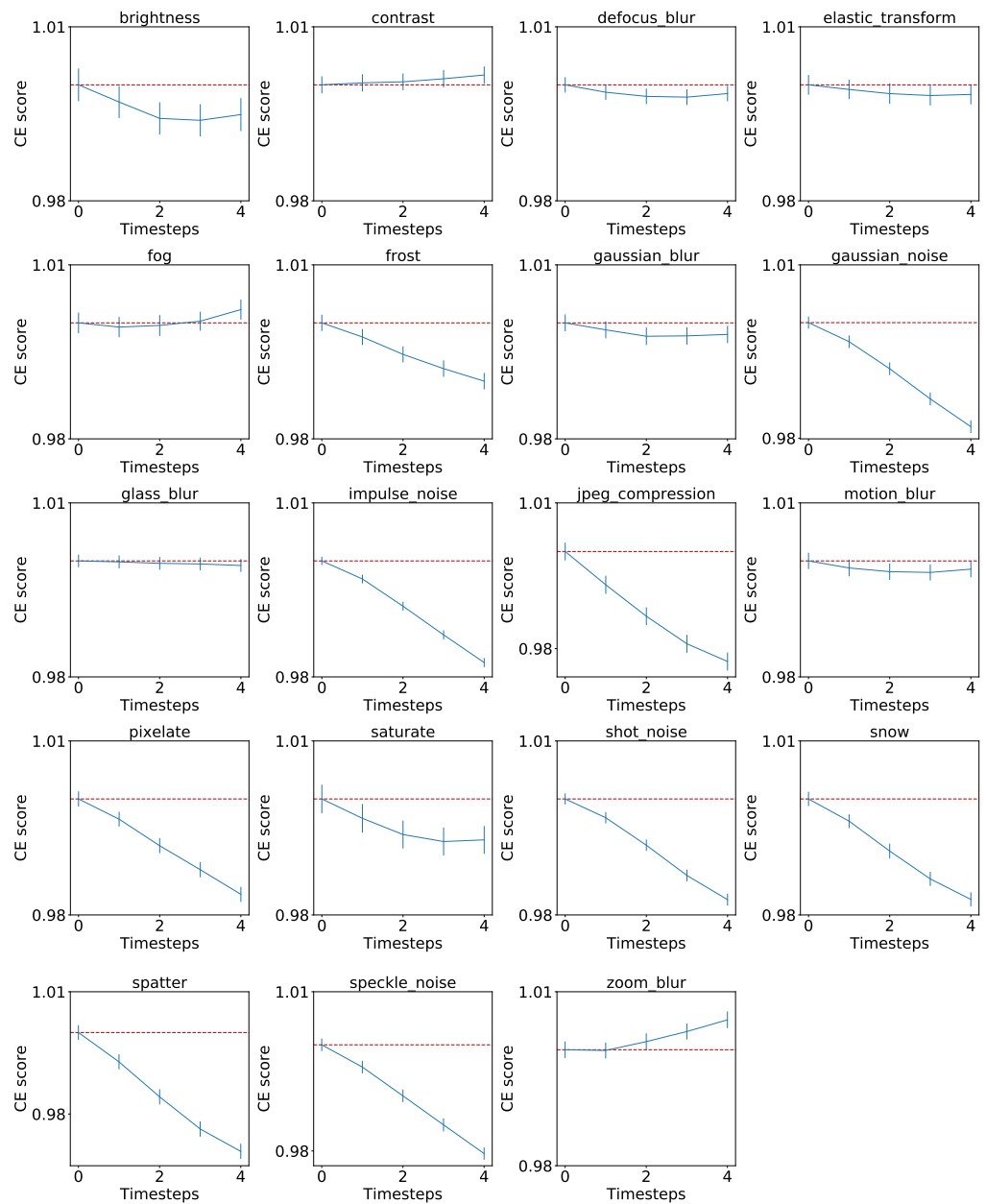

Figure S2: **PVGG16 (optimised) Corruption Error (CE) scores for all distortions:** The panel shows the CE scores calculated on the distorted images provided in the ImageNet-C dataset. The values are normalized with the CE score obtained for the feedforward VGG. The error bars denote the standard deviation of the means obtained from bootstrapping (resampling multiple binary populations across all severities.)

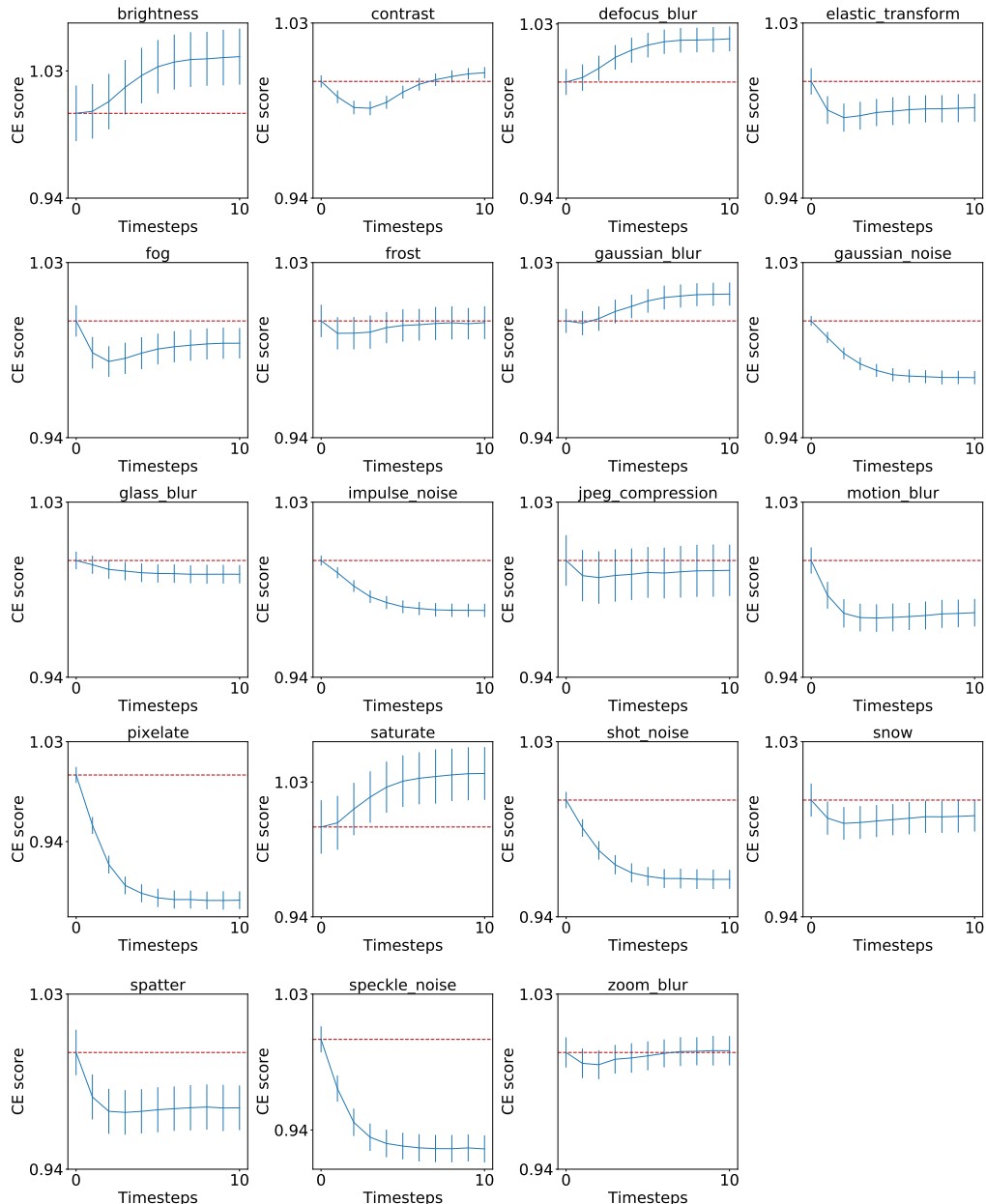

Figure S3: **PEfficientNetB0 (optimised) Corruption Error (CE) scores for all distortions:** The panel shows the CE scores calculated on the distorted images provided in the ImageNet-C dataset. The values are normalized with the CE score obtained for the feedforward EfficientNetB0. The error bars denote the standard deviation of the means obtained from bootstrapping (resampling multiple binary populations across all severities.)

### A.8 mCE scores of the optimized networks using AlexNet as a baseline

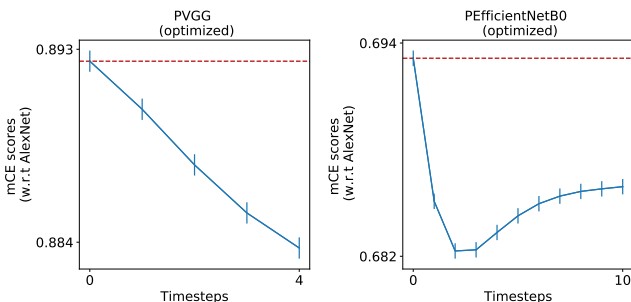

Figure S4: The mCE scores of the optimized networks (as shown in Figure 3) normalized using the score of the AlexNet network. Instead of normalizing using the score for the feedforward version of our recurrent network, to facilitate comparison with other works, we here normalize the scores using the score obtained for AlexNet network.

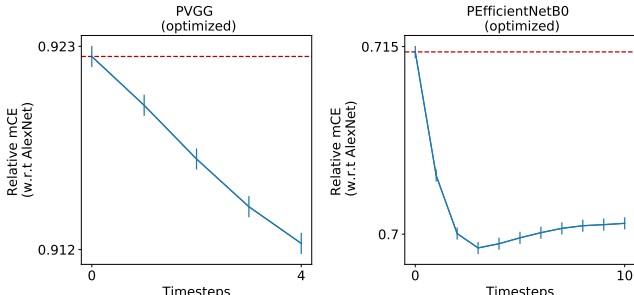

Figure S5: The Relative mCE scores of the optimized networks (as shown in Figure 3) normalized using the score of the AlexNet network. As suggested by [8], we use Relative mCE score which accounts for the changing baseline accuracy on the clean images over timesteps.

### A.9 mCE scores of a predified robust network

We also incorporated our recurrent dynamics in an already robust PEfficientNet network. As a simple approach, we just used the hyperparameters ($\alpha$, $\beta$ and $\lambda$) that were optimized for the non-robust version of PEfficientNEtB0 (on 0.25 gaussian noise) and measured its robustness against the corruptions in ImageNet-C dataset. We observed that the proposed predictive coding dynamics further helped in improving the robustness of this already robust network.

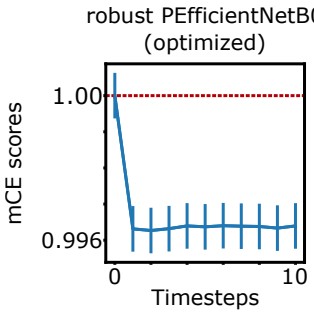

Figure S6: mCE scores of a predified version of an already robust PEfficientNetB0

## A.10 Original data for Adversarial Attacks

We provide here the non-baseline corrected versions of the data presented for adversarial attacks in Figure 4. The panels below show the success rate of the targeted attacks across timesteps calculated on 1000 images. The perturbations allowed ($\epsilon$) and the type of attack are denoted at the top.

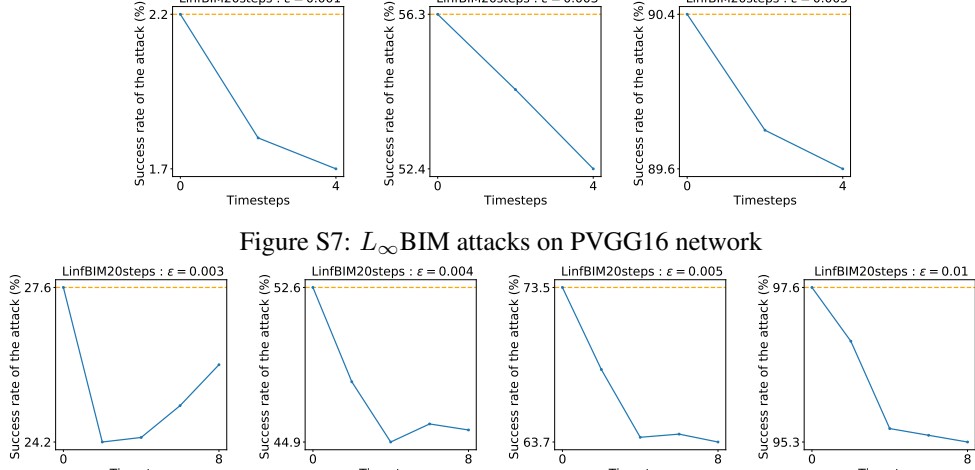

Figure S7: $L_\infty$BIM attacks on PVGG16 network

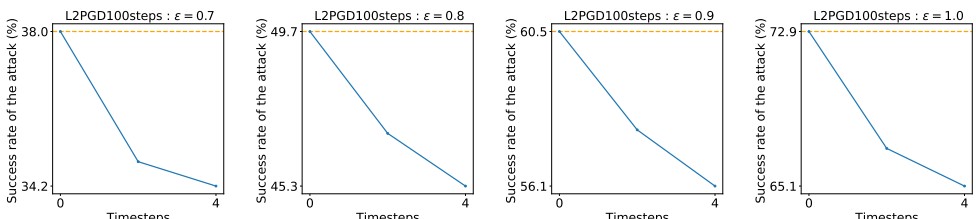

Figure S8: $L_\infty$BIM attacks on PEfficientNetB0 network

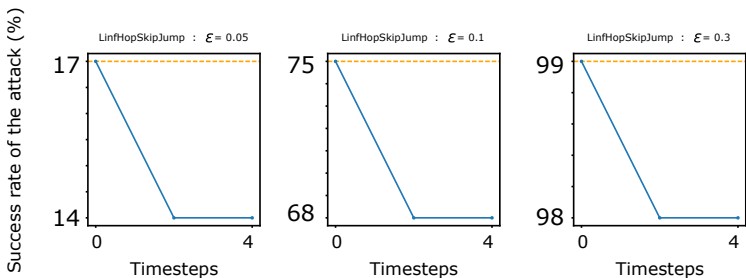

Figure S9: $L_2$RPGD attacks on PEfficientNetB0 network

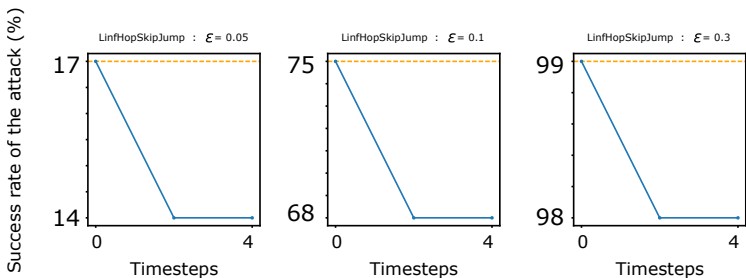

Figure S10: $L_\infty$ HopSkipJump attacks on PEfficientNetB0

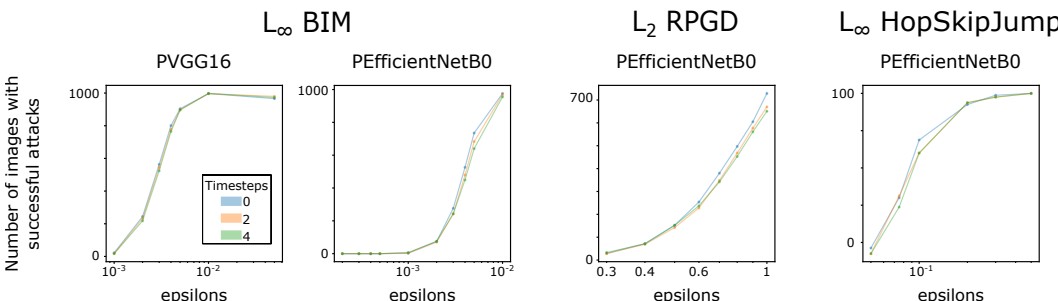

Figure S11: Adversarial Attacks with respect to epsilons. Here we show the number of successful attacks on 1000 (100 for HopSkipJump) images. Increasing the size of the epsilon leads to increase in the success rate of the attack as expected. As predictive coding timesteps increase, the curves shift slightly to the right, meaning that a slightly larger perturbation is required to fool the network. This robustness is more easily seen on Figure 4, where $\epsilon$ values are sampled near each curve's inflection point.

### A.11 Absolute values of the plots shown in the main text

| Noise Level | PVGG16 | | PEfficientNetB0 | |
|---|---|---|---|---|
| | Accuracy at t=0 | Accuracy at t=15 | Accuracy at t=0 | Accuracy at t=15 |
| $\sigma = 0.00$ | 71.63 | 71.47 | 77.29 | 75.35 |
| $\sigma = 0.50$ | 35.61 | 38.59 | 57.66 | 56.24 |
| $\sigma = 0.75$ | 16.69 | 18.46 | 37.11 | 41.05 |
| $\sigma = 1.00$ | 5.59 | 7.05 | 17.03 | 23.59 |

Table S4: Accuracy on gaussian noise-corrupted images. Here we show the accuracy obtained on images corrupted using gaussian noise (at t=0) as shown in figure 2a. All the values are calculated on the corrupted versions of the ImageNet validation dataset.

| Noise Level | PVGG16 | | PEfficientNetB0 | |
|---|---|---|---|---|
| | MSE at t=0 | MSE at t=15 | MSE at t=0 | MSE at t=15 |
| $\sigma = 0.00$ | 0.224 | 0.220 | 0.186 | 0.184 |
| $\sigma = 0.25$ | 0.342 | 0.324 | 0.223 | 0.222 |
| $\sigma = 0.50$ | 0.518 | 0.485 | 0.303 | 0.302 |
| $\sigma = 0.75$ | 0.705 | 0.660 | 0.394 | 0.392 |
| $\sigma = 1.00$ | 0.898 | 0.842 | 0.486 | 0.482 |
| $\sigma = 2.00$ | 1.689 | 1.587 | 0.848 | 0.834 |

Table S5: MSE distances for reconstructions on noisy images. Here we show the MSE distances obtained between the noisy images corrupted using gaussian noises and the reconstructions made by the models as shown in Figure 2b.

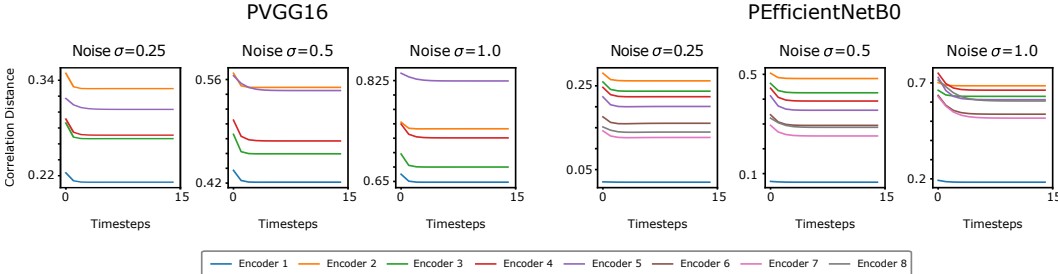

Figure S12: **Correlation distances for representations obtained on noisy images:** Here we show the absolute correlation distances obtained between clean and noisy representations as shown in Figure 2d in the main text.