# OpenReview forum: "Predify: Augmenting deep neural networks with brain-inspired predictive coding dynamics"
_NeurIPS.cc/2021/Conference — NeurIPS 2021 Poster_

### Official Review · Reviewer_NhQe · 2021-07-04

**Rating:** 7
**Confidence:** 5

**Summary:**

The authors draw inspiration from neuroscience and propose augmenting feedforward neural networks with recurrent dynamics to improve their adversarial and corruption robustness. They provide a PyTorch package that can augment any convolutional neural network with recurrence. Their empirical tests show that recurrence mostly helps with robustness.

**Limitations And Societal Impact:**

Yes

**Main Review:**

*Review summary:*

The proposed method is well explained and the provided recurrence software package could be a valuable contribution to the community. The paper could be improved by reporting more interpretable metrics and addressing inconsistencies in the results. Additionally, the literature review needs to be improved and the novelty made more explicit. I do not believe the paper is good enough for publication in its current form, but am ready to revise my score if my concerns are adequately addressed.

*Strengths:*

The paper is well written in terms of grammar and fluency. The introductory paragraphs provide good motivation for exploring recurrence in DNNs, and the model description is clear and concise. The concluding and broader impact statements are also well argued. The breadth of experiments (denoising, corruption robustness, and adversarial robustness) is helpful for providing a holistic understanding for how recurrence influences resilience to perturbations. While I didn’t look at the code itself, I believe that the provided codebase could be helpful for important future work in this space. I also appreciate the detailed comparisons to [22] and [24] in the Appendix.

*Weaknesses:*

This is a fairly long list that is not in order of importance (sorry). However, I hope many of these issues can be easily resolved by the authors.

a) I recommend additional background discussion on related prior work. There is a wealth of research attempting to use predictive coding ideas to improve DNN robustness. If we only focus on recurrence then the scope broadens further {2, 3, 8}. Although note that {8} heavily draws on the concept of analysis-by-synthesis, which is also a core idea in predictive coding. A more related example would be sparse coding, which is a predictive coding model that has been used for denoising for decades {6, 7}. Recently, several papers have also demonstrated adversarial robustness with sparse coding {1, 4, 5}. I have provided some examples here, but I would encourage the authors to look at citations of and within them to inform a proper discussion on the context of their work such that readers can properly assess the novelty of their ideas. An additional (minor) issue is that the existing references should also be updated. Many papers cited as arxiv preprints are now published in peer-reviewed venues.

b) It would be easier to compare the networks (VGG vs EfficientNet) if the authors used shared vertical axes in adjacent plots. This recommendation applies to almost every subfigure. The study’s goal is clearly not to compare VGG against EfficientNet, but understanding the relative differences between baselines can give better context for the change in performance when recurrence is added.

c) It is not clear to me what motivated the stopping criteria for the recurrence in any given experiment. The authors appear to arbitrarily stop the model at 4, 8, 10, 15, and 20 time steps. Why is this? I would like the authors to extend the time steps as long as necessary (given computational constraints) to give a complete picture of how recurrence influences the representations. Presumably all of the curves will eventually converge to a steady state -- when is that? If not, why? In a few instances (e.g. figure 4 for eps=0.003 and figure 3(b) for PEfficentNet), the performance does a reversal, which should at least be discussed.

d) When interpreting the gaussian noise experiment results (figure 2) the authors state that “Both networks demonstrate significant accuracy improvement across timesteps under noisy conditions, while maintaining a performance close to the feedforward level for clean images.” I do not believe this statement is supported by the data. For one, the use of the word “significant” implies that they did some sort of significance test, which they did not as far as I can tell. Perhaps more importantly is that for certain settings (e.g. the blue and orange curves in figure 2(a)) the recurrence appears to hurt performance. Due to the axis rescaling (see point b above), it is hard to see whether these differences matter. Finally, they should note what data that accuracy was computed on -- I assume one of the ImageNet test sets?

e) I do not understand the metric used in figure 2(b). I believe a normalized MSE distance of 1.0 indicates that the reconstruction quality with recurrence matches that without. The authors should be explicit about this. This metric is not as helpful to understand as reporting the raw MSE values (this is a theme across my suggestions). I find it interesting that there is not a consistent correlation or trend when comparing MSE against noise level. In fact, the PVGG net and PEfficientNet appear to have opposite trends, where the noise free case has the highest score for PVGG and almost lowest for PEfficientNet. There is a similar inconsistency (and similar metric confusion) in figure 2(d), where there is not a monotonic relationship between layer and distance, as I would expect. I would appreciate it if the authors would comment on this.

f) I think it would be helpful if the authors used the same images for the two networks in figure2(C) so that readers can understand the relative difference in the baselines used.

g) In terms of the corruption experiments, given that the clean accuracy has changed I think it makes more sense to use the relative CE score from [8] than the regular CE. Also, the authors do not say if they are using the same AlexNet baseline as was used in [8]. At timestep 0 they always report CE of 1.0, which leads me to guess that they are using the standard VGG or EfficientNet score as the denominator. At a minimum the authors should be explicit about how they are computing the metric instead of deferring to the citation (unless they are indeed computing the score exactly as was done in [8]). Ideally they should use AlexNet as their denominator so that the reader can properly compare their performance with other works in the field. I appreciate that the authors intended to focus on the added value of feedback, instead of achieving SOTA on some benchmark. However, it is still helpful to understand the added value if we have a good relative context.

h) The authors alternate between showing individual CE scores without hyperparameter tuning in figure 3.(a) and mCE with tuning in figure 3.(b). But in the appendix figure 6 we can see that there are tuned CE scores that perform worse with recurrence (e.g. zoom & fog for VGG). To clarify the results I recommend that the authors provide more information in the main paper. Specifically, they should acknowledge negative CE results from the tuned/untuned models in the main paper. And if the goal is to compare the results “with tuning” against “without tuning”, then they should use the same metric in both scenarios.

i) The robustness metrics used are also not standard in that field. The appendix figures 8, 9, and 10 are much more in line with what is typically reported. I don’t see the added benefit of the presentation in figure 4 when we can easily see the “baseline corrected” score by comparing time step 0 with those that are > 0 (this argument applies to many of the above points as well). Again, even though the authors wish to stress a relative comparison it is important that they report their results in such a way that it can be viewed in the context of other work. As a point of comparison, the results reported in {9} also used FoolBox and are more aligned to typical setup and formatting.

j) I’m not sure if the concerns raised in {4, 8} about backpropagating through recurrence are relevant here, but it would be helpful if the authors would explicitly say that the white-box attacks did not suffer from gradient obfuscation issues (see {9}). Even better would be if they additionally used gradient-free attacks as in {8}.

{1} Sulam et al., Adversarial Robustness of Supervised Sparse Coding, https://proceedings.neurips.cc/paper/2020/hash/170f6aa36530c364b77ddf83a84e7351-Abstract.html

{2} Krotov et al., Dense associative memory is robust to adversarial inputs, https://direct.mit.edu/neco/article-abstract/30/12/3151/8426/Dense-Associative-Memory-Is-Robust-to-Adversarial?redirectedFrom=fulltext

{3} Frosst et al., DARCC: Detecting Adversaries by Reconstruction from Class Conditional Capsules, ​​https://arxiv.org/abs/1811.06969

{4} Paiton et al., Selectivity and robustness of sparse coding networks, https://jov.arvojournals.org/article.aspx?articleid=2772000

{5} Kim et al., Modeling Biological Immunity to Adversarial Examples, https://openaccess.thecvf.com/content_CVPR_2020/html/Kim_Modeling_Biological_Immunity_to_Adversarial_Examples_CVPR_2020_paper.html

{6} Lu et al., Sparse coding for image denoising using spike and slab prior, https://www.sciencedirect.com/science/article/abs/pii/S0925231212007643?via%3Dihub

{7} Oja et al., Image Feature Extraction and Denoising by Sparse Coding, https://link.springer.com/article/10.1007/s100440050021

{8} Schott et al., Towards the first adversarially robust neural network model on MNIST, https://openreview.net/forum?id=S1EHOsC9tX

{9} Athalye et al. Obfuscated Gradients Give a False Sense of Security: Circumventing Defenses to Adversarial Examples http://proceedings.mlr.press/v80/athalye18a.html

**Time Spent Reviewing:**

5

---

> ### Author Response · Authors · 2021-08-10
> **Response to Reviewer NhQe**
>
> We thank the reviewer for their meticulous and constructive suggestions. We try to address each of them below:
>
> a) We agree with the Reviewer that while we explicitly mention all DNN-based implementations of predictive coding (to our knowledge), we did not cover all existing work on recurrence and neural networks, nor did we highlight the sparse coding literature and its relation to predictive coding. We would make it a point to include all these works, as suggested by the Reviewer, in the final version of the manuscript (in case of acceptance). Apologies also for the oversight on the references, we will update them appropriately.
>
> b) We didn’t plot the absolute values since, as you rightly pointed out, we didn’t deem it useful to directly compare VGG and EfficientNet, two baselines which are very different. VGG has an accuracy of ~72% on ImageNet whereas EfficientNet has ~77%. Adding to that is the different parametric sizes, and more importantly, different number of pcoders in their predified versions. But, as you suggested, absolute values might help interested people, and we will be happy to include those in the final version of the manuscript.
>
> c) The stopping criteria for recurrence in most of our experiments was mostly driven by our computational limitations (though we made sure in most cases that it was sufficient for observing a converging trend). Given its parametric size, generally speaking, any experiment on PVGG took a lot longer. For adversarial attacks, computational costs were high for both the networks, and hence we just demonstrated for a few selected timesteps. Otherwise, for experiments where computational constraint was not the biggest limitation, we arbitrarily chose a number of timesteps that demonstrated the convergence of the networks without having plots with long uninformative horizontal lines. We will definitely take up your suggestion and update the plots while also explicitly discussing any non-converging plots in the final version of the manuscript.
>
>
> d) We apologize for this misleading sentence. We will update the sentence in the main manuscript as follows :
> 	  “ Both networks demonstrate accuracy improvement across timesteps under high noise conditions, while maintaining a performance close to the feedforward level for clean images. ”
>
> For Figure 2(a), we don’t rescale the axes, and just plot the (Accuracy(t) - Accuracy(t=0)), implying that, for example, for PEfficientNet, in a best case scenario, predictive coding dynamics help to regain around 6 percent accuracy on noisy images compared to its feedforward counterpart. We will clarify this more explicitly and also include absolute values in the revised version of the manuscript. The Reviewer is right in understanding that the data used was images from the ImageNet validation set.
>
>
> e) Apologies for this confusion. At t=0, we pass the images through the networks’ feedback weights and get the reconstructions from the model. Indeed, there is no influence of the predictive coding dynamics at this stage. Since the absolute values of reconstruction errors don’t allow one to draw any informative inferences apart from the quality of the feedback weights, we used this value at t=0 to normalize all the subsequent error values to highlight the effect of the dynamics. That is for Figure 2(b) we plot,
>
> $$ Value\\_at\\_time\\_t =  \frac{ MSE(reconstruction\\_at\\_time\\_t, clean\\_image) }{MSE(reconstruction\\_at\\_time\\_0, clean\\_image)}  $$
>
> Hence, the figure 2(b) shows the ‘relative improvement in mean squared error’ over timesteps.
>
> For measuring the correlation distances in Figure 2(d), we use a similar rationale, since absolute values of correlation distances hardly signify anything. The exact calculation is as follows :
>
> $$ Value\\_at\\_time\\_t =  \frac{CorrDistance( noisy\\_representation\\_at\\_time\\_t , clean\\_representation\\_at\\_time\\_t )}{ CorrDistance (noisy\\_representation\\_at\\_time\\_0 , clean\\_representation\\_at\\_time\\_0 )} $$
>
> We will further clarify our explanation for these metrics,  and also include absolute values in the final version of the paper in order to alleviate any possible confusion.
>
> Each pcoder will have its unique efficacy for converging to an optimal point given the equilibrium reached by the overall network. Since this will depend on a multitude of pcoder-specific factors like the quality of feedback weights, the values of the hyperparameters, etc., we believe it is not unexpected to find no consistent trend across pcoders and networks.
>
>
> f) We used the different images just to increase the variety in our reconstructions. We will be happy to update these images according to the reviewer’s suggestion in the final accepted version of the manuscript.
>
> g) We appreciate the suggestion from the Reviewer and will include a plot for Relative CE score in the final version of the paper. The Reviewer is right in assuming that we use the CE (or mCE) scores of each corresponding feedforward model as a baseline (denominator). This was mostly to maintain consistency in scale across all our plots, and we note that it does not violate the recommendations from [8]. If our paper is accepted, in the final version we will clarify the calculation of this metric while also including a plot with AlexNet as the baseline, as recommended by the reviewer.
>
> h) We apologize for this confusion. Indeed the goal of Figure 3 was to demonstrate noise robustness using the predictive coding dynamics. Though the untuned network showed robustness against a variety of noises, the overall mCE scores didn’t improve until we used tuned hyperparameters. Similarly, tuned hyperparameters didn’t always imply that the network gained additional robustness against all types of corruptions. As recommended, we will expand on these negative cases in our next revision of the manuscript.
>
> i) We agree with the reviewer that the presentation choice is not standard in the field. Again, as noted by the reviewer themselves, we had compared the robustness with the feedforward counterparts to maintain consistency throughout our article. But in case of acceptance, we will revise the main manuscript to include the plots in the format of supplementary figures 8, 9, 10, as suggested by the reviewer.
>
> j) We are currently performing gradient-free attacks on the networks, as recommended, and intend to include the corresponding results in the final manuscript.

---

> > ### Comment · Reviewer_NhQe · 2021-08-16
> > **Concerns mostly addressed -- a few minor comments**
> >
> > Thank you for taking the time to consider my suggestions. Overall I think the quality of the paper will be improved when accounting for the promised updates after my own and the other reviews. However, I do still have a few clarifications below that I would like the authors to consider. I hope that these will be minor points and the discussion can conclude quickly.
> >
> > a) I appreciate the care to include this related context. Note also that at least one of the sparse coding models for robustness, {5}, could be considered a “DNN” and I believe related work from that group also fits a “DNN” classification. I trust the authors will take care in framing the sparse coding literature in the context of their contribution.
> >
> > b) I think including figures with absolute values in the appendix with a reference to it in the main manuscript would be helpful. See my note below about axis rescaling in general.
> >
> > c) Please include this explanation in the main manuscript, and if possible make an effort to have some sort of consistent criteria for stopping the recurrence, even if that criteria changes per experiment type. Even with your response, I still don’t understand a few stopping criteria beyond “arbitrary.” For example, the apparent convergence of PVGG16 and PEfficientNetB0 in figure 3(b) is totally different. EfficientNet seems to do a reversal, and is possibly reaching a plateau. However, VGG is continuing to decline and might show a similar reversal with more inference steps. You mentioned that VGG is easier to run, so why stop early? Maybe I’m reading these incorrectly, if so please clarify. Another example would be figures 2(a) and 2(b). I am guessing that both of those metrics can be computed simultaneously without additional cost, so why were you able to run 20 time steps for EfficientNet when computing accuracy but only 15 when computing the MSE distances, and of course the same argument applies for VGG when comparing the two metrics.
> >
> > d) I think your revised sentence is more accurate and clear. I would appreciate an attempt to explain why the accuracy does not consistently improve in low-noise settings, or at least an acknowledgement of it, but I understand that this might have to be left as an open question. By “axis rescaling”, I mean a physical distance on the y axis corresponds to a different Improvement in Accuracy between the two subplots. To say it another way, the distance from 0 to 3 in the PVGG subplot is about the same as the distance from 0 to 6 in the PEfficientNet plot. See my note below for a longer discussion.
> >
> > e) Thank you for the clarification, this makes sense and seems reasonable. Including the absolute values in the appendix would be sufficient. I guess the answer to my followup question will have to be relegated to future work, which is acceptable to me.
> >
> > f) Additional images could be put in the appendix if one wanted to show more variety.
> >
> > g-j) Thank you, I appreciate the clarifications. Please take care to justify when you are using CD vs. mCE, and I look forward to seeing the gradient-free attack results.
> >
> > Additional note: I should have been more clear about my general point with the axis rescaling. My suggestion is for you to equalize the y-axis values in many of your adjacent plots. For example, in figure 2D you have specific sets of subplots for PVGG16 & PEfficientNetB0 and within those subplots your y-axis values are adjusted to show a relative minimum. There are other examples in the responses above. This makes it difficult to observe overall trends, and can over-exaggerate perceived differences. I suggest that you look through each grouping of subplots carefully and assess if they really _need_ to have different (unshared) y-axes. Without a justification the default action should be to share the axes.

---

> > > ### Comment · Reviewer_NhQe · 2021-09-01
> > > **Awaiting confirmation**
> > >
> > > We are quickly approaching the discussion deadline. Would you mind confirming that you received and understand the final requests in my above comment? I do not believe it'll be difficult for you to make those changes, but some confirmation that they will be done would raise confidence in general for the final assessment.
> > >
> > > Thank you

---

> > > > ### Author Response · Authors · 2021-09-01
> > > > **Confirmation of the changes with additional results**
> > > >
> > > > Yes, we confirm that we received and understood the Reviewer’s final requests, and intend to update the final paper accordingly. We would like to apologise to the Reviewer for the delay in our response--we were finalizing simulations, and waiting to get higher numbers of runs on a few experiments to share our results with the Reviewers.
> > > >
> > > > Regarding the general point you made about axis rescaling, we agree with you that it is not always necessary to have distinct axes between plots of PVGG and PEfficientNet. We have included a version of the plots 2A and 2B with shared axes in an anonymized google document for your perusal:
> > > > [Figure 2A](  https://drive.google.com/file/d/1BE4MGC2ovEzpq1Cd0fnYv1ct-Y3CswtI/view?usp=sharing)
> > > > [Figure 2B](https://drive.google.com/file/d/1mHT_DQfIwVIzjI1GmsSMcMJar2RQlSCF/view?usp=sharing)
> > > >
> > > > We hope this can  address the general concern you had regarding our choice of axes. We intend to put these plots in the final version of the manuscript.
> > > >
> > > >
> > > > We have also included the results for gradient-free adversarial attacks at - [Gradient-free Adversarial Attacks](
> > > > https://drive.google.com/file/d/1DSu4JEut6TBJprB8IG8xhQlLQKnDWvac/view?usp=sharing)
> > > >
> > > > For these experiments, we decided to use a query-efficient decision-based attack known as HopSkipJump {1}, which does not use any information regarding network architecture but solely its output decisions. Thus without using any gradient information, the attack tries to get closer to the decision boundary of the network. With the same setting as described in the paper (targeted attacks with same targets for all timesteps), we find adversarial perturbations against the PEfficientNetB0 network in the  $L_\{inf}$ norm. In the figure, we show the data for 50 images (the number that this computationally intensive process allowed us to reach up to now), and intend to keep increasing this number until the final deadline for the manuscript. We observe that, even against this non-gradient-based attack, the robustness of the network improves over timesteps, suggesting that it is likely due to the ability of the dynamics to project towards the manifold, and not due to gradient masking. Note that, to facilitate meaningful comparisons for the Reviewers with the figures already present in the paper, we have kept the same format as used in the paper-- but intend to update them further after including the suggestions provided by the Reviewer.
> > > >
> > > >
> > > > We now address all the other points raised by the Reviewer below :
> > > >
> > > > a) - Thanks a lot for these references. We would be careful to include these while also thoroughly framing the sparse coding literature in the final version of the paper.
> > > >
> > > > c) - We would like to clarify that for ImageNet-C, PVGG is harder to run than PEfficientNet due to its larger parametric size. For example, for Figure 3B, which has 19 different corruptions with 5 severities, it requires evaluation on an ImageNet validation set 95 times. This task becomes computationally prohibitive, the reason why we stopped the PVGG at 4 timesteps. Indeed, the Reviewer is right in mentioning that PVGG seems to help beyond 4 timesteps, something that would have helped us to further highlight the benefits of our dynamics. At the same time, for Figure 2A and 2B (Gaussian noise), the experiments are not that computationally expensive. Given that we have a few more weeks if we are accepted, we intend to get more timesteps for both the networks.
> > > >
> > > > b,d,e,f,g,h,i,j) - Thanks for your comments, we will be careful to clarify when we use CE and when we use mCE. We will also show the absolute values as per your suggestion in the appendix.
> > > >
> > > > We hope these additional experiments and the clarifications will reinforce your confidence in the contributions of our work.
> > > >
> > > > {1} Chen, J., Jordan, M. I., & Wainwright, M. J. (2019). Hopskipjumpattack: A query-efficient decision-based attack. arXiv 2019. arXiv preprint arXiv:1904.02144, 3.

---

> > > > > ### Comment · Reviewer_NhQe · 2021-09-02
> > > > > **Increased score**
> > > > >
> > > > > Thank you again for your reply. The provided figures and promised updates for myself as well as the other reviewers is sufficient for me to consider this paper worthy of publication. I have increased my score two points.

---

### Official Review · Reviewer_WCib · 2021-07-16

**Rating:** 6
**Confidence:** 5

**Summary:**

This paper implemented the predictive coding dynamics into deep neural nets, and observed the increase in robustness against various corruptions. The authors also provided a pytorch-based package for easy implementation of the proposed dynamics into deep neural nets.

**Limitations And Societal Impact:**

The authors have provided sufficient discussion on this.

**Main Review:**

The predictive coding principle has gained increasing attention in the machine learning community. Despite not being the first to incorporate the predictive coding dynamics into neural networks, this paper did a good job in formulating the predictive coding principle in a clean way and they were able to test out the ideas on large scale datasets like Imagenet. The paper was well written and provided enough details to reproduce their results. In general, I enjoyed reading this paper.
However, I do have some comments and questions regarding the proposed mechanism as well:
1) In Eq. 2, the feedforward representations were updated to reduce the reconstruction error in the lower level (the alpha term). However, if we believe that the feedforward representations for noisy images are far away from that of the clean images and will cause wrong classification, why reducing the distance between the feedback representation and this "wrong" forward representation helps robustness? I understand that experimentally we see some positive results, but it would be good to comment on this point.
Also, it may be good to take a pretrained robust feedfoward network (for example, network trained with data augmentation techniques or adversarial training), and see whether the proposed mechanism can still improve the robustness.
2) In the paper, the authors only reported relative improvements, but I am curious to see the absolute value of robustness accuracy. Since the authors take pretrained networks, I am wondering if similar robustness improvements can also be achieved by training the forward model differently (for example, using data augmentation or adversarial training). This is related to the last point I mentioned in 1). It would be good to show whether feedback can further improve robustly trained feedforward models.
3) Since the authors introduced more weights in the model (the feedback path), could the robustness improvement be a result of using larger model? It would be good to take a deeper or wider feedforward model to match the number of parameters in the feedback model and see if the improvement is still there.
4) In the pseudo code, it looks like the authors loop through all PCoder modules first, and then move on to the next time step. How would this compared with performing several update steps first until one PCoder is convergent and then move on to the PCoder in the next level?
5) For the adversarial robustness experiment, I was wondering how the authors perform the attack? Are they performing end-to-end attack considering the predictive coding dynamics? If not, this should be evaluated. In addition, when performing end-to-end attack, adversarial attack may suffer from gradient obfuscation problem, so it would also be good to add black-box attack evaluation.
6) How sensitive is the method's performance with respective to the coefficient \alpha in Eq. 2? In Table 2 and Table 3 in appendix, \alpha is fixed to be 0.01. But this term is an very important term for the predictive coding principle. Also the authors scaled the gradient based on feature size, so it seems that this is a sensitive hyper-parameter.


**Time Spent Reviewing:**

5 hours

---

> ### Author Response · Authors · 2021-08-10
> **Response to Reviewer WCib**
>
> We are extremely grateful to the Reviewer for their insightful comments.
>
> 1) The Reviewer has indeed highlighted an interesting question. Predictive coding can be understood as a Bayesian framework, with feedback ($\lambda$) acting as a proxy for the *prior* and the gradient correction ($\alpha$) and feedforward ($\beta$) as a proxy for *evidence*. Every task involves reaching some form of compromise between priors and evidence. As the reviewer points out, in the presence of input noise, relying heavily on wrong evidence will definitely hamper the performance, so the hope in that case is that the network may counter this by relying more on the prior.
>
>     Although in this paper we aimed to establish our recurrent dynamics in a hierarchical network, and investigate the resulting network properties, especially in the context of robustness, in another submission to the conference we started with a similar network and asked another important question, related to the reviewer’s point — how does the network adapt its hyperparameters (controlling the feedforward, feedback and error correction terms) in the presence of noise? This paper can be found at  https://openreview.net/forum?id=r9meDe5kwVn  .
>
>     In one of the experiments of that companion paper we ablated various hyperparameters (Figure 2C) in the equation and thus measured their impact on performance.  We found that using the $\alpha$ term without feedback error ($\lambda=0$) downgrades the network’s performance in the presence of Gaussian and salt-and-pepper noise, consistent with the reviewer’s suggestion. This confirms the importance of the other terms (feedback or ‘prior’) for correcting the representations in the presence of input noise.
>
>     On the other hand, predictive coding is a very generic framework, and apart from noise robustness, there are likely many other tasks (that we don’t cover in this work) for which emphasizing the evidence (through the $\alpha$ error correction term) could be beneficial. This is why it was important for us to include this term in our equations and in our “Predify” package.
>
> 2) Since we envisioned the main goal of the paper to demonstrate the effectiveness of the predictive coding dynamics on (the representations of) a pretrained feedforward model, we thought that the corresponding feedforward models would act as a meaningful baseline to interpret our results. As is also suggested by another Reviewer, we will be glad to add the absolute accuracy values in the final version of the manuscript.
>
>     Definitely any attempt at achieving state-of-the-art robustness would require careful tuning of the hyperparameters and appropriate training of both feedforward and feedback weights, and this is why we have been careful to be conservative in our claims. We agree that by directly training on out-of-distribution samples, using data augmentation or adversarial training, one could definitely achieve higher robustness. However, we also note that the associated computational requirements (especially for adversarial training) can be prohibitive, and so we have left this question open for future work. As per your suggestion, we are working on predify-ing a robust model, and will aim to provide the corresponding results for the final submission, if the paper is accepted.
>
>
> 3) In this paper we have used only standard feedforward models available in the literature, and did not train the feedback weights (extra parameters) for the classification objective itself, but rather on a very separate reconstruction objective (unsupervised training).
>
>     Importantly, training a different model, either a control deeper architecture or even the same architecture with a different random seed will imply learning a different representation space. We believe that the benefits of our dynamics are best demonstrated by testing on top of ‘an already learned’ and fixed representation space, robust or not. Hence, instead of using deeper models as control, as you suggested above, using a robust model and augmenting it with our recurrent dynamics would be more meaningful.
>
>     On the other hand, when both feedforward and feedback weights are trained on the same classification objective, parameter number becomes an important control. While addressing other questions in our companion paper, in figure 2B, we did include control models that matched the parametric sizes and showed that the models implementing the recurrent dynamics performed on par or better than these control models.
>
>
>
> 4) The suggestion you provide is quite rational, and indeed has been tried before by other implementations {1,2}. There were two reasons we didn’t implement our dynamics in this fashion. Firstly, due to biological inspiration: We know that the brain performs a quick feedforward inference of the object it's viewing, and that this inference is then modified/finetuned using recurrence, which is especially helpful for challenging inputs {3,4} . Implementing a network where each layer in the hierarchy waits for its lower layer to converge would be like a brain where the amygdala must wait several hundred milliseconds to confirm the presence of a tiger before giving its flight or fright response. We didn’t consider this approach as it seemed biologically implausible.
>
>     Secondly, this was also because of our interpretation of the recurrent dynamics. The interplay between feedforward and feedback drives should allow each layer to reach an equilibrium. If the first pcoder converges without having received any changing “adaptive” feedback from the second pcoder, the dynamics cannot include any meaningful feedback.
>
> 5) Yes, the reviewer is correct in their understanding. We unroll the whole network through time to perform the adversarial attacks. We also intend to perform some non gradient-based attacks and add them to the final version of the manuscript.
>
> 6)  As discussed before (point 1 above), a strong $\alpha$ (emphasizing the evidence) can prove detrimental to performance in the presence of noise. However, the $\alpha$ term should not be considered in isolation since it is highly affected by the other terms. As highlighted in our companion work on the role of hyper-parameters, the value and sensitivity of the $\alpha$ term can vary a lot from one network to the other or even across different PCoders in a single network.
>
> {1} :  Han, Kuan, et al. "Deep predictive coding network with local recurrent processing for object recognition." arXiv preprint arXiv:1805.07526 (2018).
>
> {2} : Wen, Haiguang, et al. "Deep predictive coding network for object recognition." International Conference on Machine Learning. PMLR, 2018.
>
> {3} : Kar, Kohitij, and James J. DiCarlo. "Fast recurrent processing via ventrolateral prefrontal cortex is needed by the primate ventral stream for robust core visual object recognition." Neuron 109.1 (2021): 164-176.
>
> {4} : Kar, Kohitij, et al. "Evidence that recurrent circuits are critical to the ventral stream’s execution of core object recognition behavior." Nature neuroscience 22.6 (2019): 974-983.

---

> > ### Author Response · Authors · 2021-09-01
> > **Additional experiments suggested by the Reviewer**
> >
> > We would like to provide further experiments which the Reviewer had suggested, to help address their concerns.
> >
> > One of the suggestions provided by Reviewer (see point 3 above) was to test the ability of our dynamics to improve the robustness of an already robust network. For this, we started by predifying a robust version of the EfficientNetB0, made publicly available by (https://arxiv.org/pdf/1911.09665.pdf). This network is already better (than its corresponding vanilla EfficientNetB0) on the ImageNet-C dataset as can be seen in the Table1 of the original paper.
> >
> > We first calculated the mCE scores of the predified robust EfficientNetB0 (hereafter called “robust PEfficientNetB0”) using the default hyperparameters used in our paper (beta=0.8, lambda=0.1, alpha=0.01); like its non-robust counterpart (PEfficientNetB0), we observed that these hyperparameters did not aid to reliably  improve the mCE scores over the iterations (data not shown). As discussed in the main paper, we also had hyperparameters that were optimized on gaussian noises of 0.25 and 0.5 for the non-robust model. We thus used the exact same hyperparameters (optimized for a non-robust network) on this robust version of the PEfficientNetB0 and observed that they do further help in improving the mCE scores of the robust model. The mCE scores  can be viewed at :
> > [Parameters optimized for 0.25 gaussian noise](https://drive.google.com/file/d/1CtlZi5z5YIpj1Uvx6P4cuIxc_lLNXn3n/view?usp=sharing)
> > [Parameters optimized for 0.5 gaussian noise](https://drive.google.com/file/d/1Wwdkjfjadlodg-Z8SwnmwXu0oXJX-ZwL/view?usp=sharing)
> >
> > Similarly, to address the concern of gradient obfuscation (point 5 above), we also performed additional experiments against gradient-free attacks, the results of which can be viewed at - [Gradient-free Adversarial Attacks](https://drive.google.com/file/d/1DSu4JEut6TBJprB8IG8xhQlLQKnDWvac/view?usp=sharing)
> >
> > For these experiments, we decided to use a query-efficient decision-based attack known as HopSkipJump {1}, which does not use any information regarding network architecture but solely its output decisions. Thus without using any gradient information, the attack tries to get closer to the decision boundary of the network. With the same setting as described in the paper (targeted attacks with same targets for all timesteps), we find adversarial perturbations against the PEfficientNetB0 network in the  $L_\{inf}$ norm. In the figure, we show the data for 50 images (the number that this computationally intensive process allowed us to reach up to now), and intend to keep increasing this number until the final deadline for the manuscript. We observe that, even against this non-gradient-based attack, the robustness of the network improves over timesteps, suggesting that it is likely due to the ability of the dynamics to project towards the manifold, and not due to gradient masking.
> >
> > We hope that these experiments, together with the previous clarifications, will help reinforce the Reviewer's confidence in the contributions of our work.
> >
> > {1} Chen, J., Jordan, M. I., & Wainwright, M. J. (2019). Hopskipjumpattack: A query-efficient decision-based attack. arXiv 2019. arXiv preprint arXiv:1904.02144, 3.

---

> > > ### Comment · Reviewer_WCib · 2021-09-14
> > > **Thanks for the response**
> > >
> > > I'd like to thank the authors for their careful response. I am happy to see the new experiment results. I do think the experiments make the paper more comprehensive, and the paper is above the acceptance threshold.

---

### Official Review · Reviewer_WgTo · 2021-07-16

**Rating:** 7
**Confidence:** 5

**Summary:**

The authors proposed a new algorithm to implement a popular neuroscience theory, i.e. predictive coding, in deep neural networks (DNN). The feedforward connection weights are pretrained for object recognition and the feedback connection weights are trained for image reconstruction, and then predictive coding introduces recurrent computations with the trained DNNs. The authors use two image recognition DNNs (VGG16 and EfficientNetB0) in experiments. They found that the recurrent computation brings improvement in object recognition and image/feature reconstruction over time given noisy inputs. This property of predictive coding improves DNN robustness to noisy examples and adversarial attacks. The authors also provide a python package to allow community to explore the predictive coding algorithm.

**Limitations And Societal Impact:**

Yes.

**Main Review:**

Predictive coding is an important theory in neuroscience and recently some researchers have explored several implementations with deep neural network to help understand the theory. How neurons recurrently process information is still unknown in predictive coding theory. This paper proposed a new implementation (though have some similarity to previous works) and did extensive experiments and found desirable properties of recurrent computation over time. This is very interesting and valuable to both neuroscience and deep learning communities. The writing is clear and easy to understand.

The feedforward connection weights in DNN are trained with forward propagation for object recognition (pretrained) and the feedback connect weights are trained with reconstruction of lower layer features. No recurrent computation is involved in training the connection weights, which may not be the case in the brain. It would be good if the author can discuss this in the perspectives of both neuroscience and machine learning.

**Time Spent Reviewing:**

4

---

> ### Author Response · Authors · 2021-08-10
> **Response to Reviewer WgTo**
>
> We are extremely grateful to the Reviewer for their comments and share their enthusiasm for the value this work will bring to both machine learning and neuroscience communities. We will be delighted to include the suggested discussions in the final version of the paper. With an aim to initiate an insightful discussion and take the Reviewer’s opinion, we here start by discussing our understanding below :
>
> The choice of using different objectives for the feedforward and feedback weights, at least from a machine learning perspective, was a simple one.  Firstly, training the feedback weights with recurrence requires one to unroll the network over timesteps. Hence, training a large network like PVGG for say 5 or 10 timesteps would incur significant computational challenges. Some attempts have tried training both feedforward and feedback connections together for classification {1} at the final timestep for relatively smaller networks, but as we discussed in our explorations in the Appendix we found that the resulting network ended up classifying correctly at the last timestep, with very poor performance during early timesteps. Indeed this problem can be addressed by training over time-averaged metrics, such as the average cross-entropy loss for $N$ timesteps. Secondly, our use of a one-step reconstruction objective allowed us to train the feedback weights independently of the various hyperparameters of our predictive coding dynamics ($\beta$, $\lambda$, and $\alpha$). Training these weights using recurrence would require to (i) either fix the values of these hyperparameters beforehand, leading to constraints of expensive hyperparameter explorations, or (ii) directly train these hyperparameters as parameters of the model while ensuring some well-justified constraints that don’t allow the network to reach trivial values.
>
> From the neuroscience perspective, whether and how the brain combines discriminative and generative representations has been an open question addressed by many researchers, e.g. {2,3,4}.  Our approach of a discriminative feedforward coupled with generative feedback could be considered another attempt in this direction.
>
>
> {1} : Wen, Haiguang, et al. "Deep predictive coding network for object recognition." International Conference on Machine Learning. PMLR, 2018.
>
>
> {2} : Huffman, Derek J., and Craig EL Stark. "Multivariate pattern analysis of the human medial temporal lobe revealed representationally categorical cortex and representationally agnostic hippocampus." Hippocampus 24.11 (2014): 1394-1403.
>
>
> {3} : DiCarlo, James J., et al. "How does the brain combine generative models and direct discriminative computations in high-level vision?." (2021).
>
>
> {4} : Al-Tahan, Haider, and Yalda Mohsenzadeh. "Reconstructing feedback representations in the ventral visual pathway with a generative adversarial autoencoder." PLoS Computational Biology 17.3 (2021): e1008775.

---

> > ### Comment · Reviewer_WgTo · 2021-08-31
> > **Response**
> >
> > Thanks for adding the discussion. It is very important for readers to know why the predictive coding is implemented like this. My comment is addressed after adding the discussion.

---

### Official Review · Reviewer_4h3w · 2021-07-16

**Rating:** 4
**Confidence:** 1

**Summary:**

Taking inspiration from the theory of predictive coding in neuroscience, the authors implement CNNs with layerwise predictive activity, as a form of unsupervised training. Experiments show that this approach is helpful in case of image corruption.

**Limitations And Societal Impact:**

The novelty of the work is very limited. Since the model is not new, they only show that the application brings advantages in the case of corrupted images with perturbations.

The experimental part is also limited. The authors only evaluate one type of image noise. It would have been interesting to see if this theory could contribute to improving robustness against adversarial attacks.

**Main Review:**

The paper is generally well written and easy to follow.  The illustrations are effective. The section dedicated to the description with code snapshots of the developed library is unnecessary in the paper and could be relegated to supplementary material or link documentation. I would use the space available to provide more references to the state of the art and describe the proposed approach in more detail.

The idea of predictive coding applied to neural network layers is not new. The authors should expand the related works section to be more inclusive of more recent work.

**Time Spent Reviewing:**

1

---

> ### Author Response · Authors · 2021-08-10
> **Response to Reviewer 4h3w**
>
> *“The idea of predictive coding applied to neural network layers is not new. The authors should expand the related works section to be more inclusive of more recent work.”*:
> We thank the reviewer for their comments. We feel that the reviewer  might have missed a few points from our manuscript and would like to apologise if our presentation was not optimal. We did include in the “Related Works” section all works that we know about related to predictive coding in machine learning; we have also expanded our discussion of the most related works, like [22] and [24], in the Appendix. At the same time, we would be delighted to include any more recent literature that might have been missed, if the reviewer would be kind enough to provide the corresponding reference(s).
>
> *“The experimental part is also limited. The authors only evaluate one type of image noise.”*:
> We are puzzled by the comment: even though we used only gaussian noise for discussing projection towards the manifold in Figure 2, we have also tested the model against a variety of noises, namely ImageNet-C which has 19 different types of corruptions applied at five varying levels of severities (Figure 3 in the main text). In the next revision of the paper, we will be sure to highlight this fact more prominently.
>
> *“ It would have been interesting to see if this theory could contribute to improving robustness against adversarial attacks.”*:
> We are equally puzzled by this comment since we indeed evaluated the robustness of our networks against a few adversarial attacks (Figure 4). Though not exhaustive, we believe that these explorations, along with our experiments on ImageNet-C, demonstrate the role that predictive coding  dynamics can play in improving robustness, and that they warrant further investigation. Similarly, we intend to mention our tests of adversarial attacks more prominently in the next revision, possibly as early as the Abstract.
>
> In light of these clarifications, and given the fact that all the answers to the reviewer’s criticisms already lie in the paper (and that we intend to highlight them further in the final manuscript), we hope the reviewer will consider revising their harsh rating.

---

### Decision · Program_Chairs · 2021-09-27

**Decision:**

Accept (Poster)

**Comment:**

This paper introduces a framework for incorporating recurrent feedback connections based on predictive coding principles into feed-forward networks. Overall, the reviewers expressed excitement for the integration of neuroscience with ML and thought that the evaluations were convincing. However, there was initially high spread in reviewer scores due to concern over the novelty of the model and approach. After the detailed author responses and a thorough discussion, the consensus moved towards acceptance and two reviewers increased their scores.